# Recent Advances in Steroid Discovery: Structural Diversity and Bioactivity of Marine and Terrestrial Steroids

**DOI:** 10.3390/ijms26073203

**Published:** 2025-03-30

**Authors:** Grzegorz Hajdaś, Hanna Koenig, Tomasz Pospieszny

**Affiliations:** Department of Bioactive Products, Faculty of Chemistry, Adam Mickiewicz University, Uniwersytetu Poznańskiego 8 Street, 61-614 Poznań, Poland; grzhaj@amu.edu.pl (G.H.); hanna.koenig@amu.edu.pl (H.K.)

**Keywords:** steroids, natural products, marine-derived steroids, steroid pharmacology, bioactive compounds, drug discovery, medicinal chemistry

## Abstract

Steroids have been pivotal in medicine and biology, with research into their therapeutic potential accelerating over the past few decades. This review examines recent steroid discoveries from marine and terrestrial sources, highlighting both novel compounds and those with newly identified biological activities. The structural diversity of these steroids contributes to their wide range of biological activity, including anticancer, antimicrobial, antidiabetic, anti-inflammatory, and immunomodulatory properties. Particular emphasis is placed on steroids derived from marine invertebrates, fungi, and medicinal plants, which have shown promising therapeutic potential. Advances in analytical techniques such as NMR spectroscopy and mass spectrometry have facilitated the identification of these compounds. These findings emphasize the growing importance of steroids in addressing pressing global health issues, particularly antibiotic resistance and cancer, where new therapeutic strategies are urgently needed. Although many newly identified steroids exhibit potent bioactivity, challenges remain in translating these findings into clinical therapies. Ongoing exploration of natural sources, along with the application of modern synthetic and computational methods, will be crucial in unlocking the full therapeutic potential of steroid-based compounds.

## 1. Introduction

Steroids have played a pivotal role in medicine and biology since their discovery in the early 20th century [1,2]. Their significant impact on human physiology and disease treatment has driven advancements in endocrinology, oncology, and immunology [3,4]. Initial research focused on steroid hormones like cortisone and testosterone, which transformed into therapeutic approaches [5,6,7,8,9]. Especially over the past two decades, as the need for new bioactive compounds grew, steroid research has expanded significantly to cover a wide range of sources, including marine organisms as well as terrestrial organisms. New steroid structures with unique biological properties have been discovered, broadening our understanding of the therapeutic potential of this group of compounds [10,11,12].

Recently, attention has turned to novel steroids from marine and terrestrial sources, owing to their unique biological activities [13,14]. Marine ecosystems, particularly invertebrates such as echinoderms, tunicates, and sponges, have provided many new steroids. These steroids often possess distinct structural features, such as hydroxylated side chains and sulfate groups [15,16]. Simultaneously, medicinal plants and fungi have yielded steroidal alkaloids and saponins with significant therapeutic potential [17,18,19,20,21,22]. Over the last 20 years, many studies have reported the antitumor, antimicrobial, antidiabetic, and anti-inflammatory effects of these newly discovered steroids.

This review examines some recent discoveries about steroids made in the last two decades, focusing mainly on compounds from both marine and terrestrial environments. It also explores discoveries regarding previously known steroids, emphasizing their structural diversity and properties that had not been investigated before. Emphasis is placed on their potential therapeutic uses.

## 2. Marine Organisms

### 2.1. Marine Invertebrates

Marine natural product research, especially involving invertebrates like echinoderms, has proven to be a rich source of lead compounds for drug discovery [23,24,25,26]. Echinoderms such as *Acanthaster planci* have adapted to challenging marine environments by producing a diverse range of bioactive secondary metabolites. Previous studies on *A. planci* have led to the isolation of novel steroidal structures with antimicrobial, anti-inflammatory, and cytotoxic properties [27,28,29,30].

Chemical studies on *Acanthaster planci*, commonly known as the crown of thorns starfish, led to the isolation of two new steroids: 5α-cholesta-24-en-3β,20β-diol-23-one **(1)** and 5α-cholesta-9(11)-en-3β,20β-diol **(2)** (Figure 1) [31]. Their structures were determined using spectroscopic techniques. Compound **(1)** exhibited notable antibacterial activity, with a zone of inhibition measuring 21.0 ± 0.06 mm against *Pseudomonas aeruginosa*, while compound **(2)** displayed moderate activity (13.0 ± 0.03 mm). Both compounds were also evaluated for their antitumor properties against human breast carcinoma cells (MCF-7), with LC_50_ values of 49 ± 1.6 μg/mL for compound **(1)** and 57.5 ± 1.5 μg/mL for compound **(2)**. These results are comparable to those of the commonly used chemotherapeutic drug cisplatin (LC_50_ 46 ± 1.1 μg/mL). Additionally, both steroids demonstrated anti-diabetic activity by inhibiting α-glucosidase, with IC_50_ values of 58 ± 0.8 μg/mL and 55 ± 0.5 μg/mL, respectively [31].

Recent studies have also demonstrated the presence of the cytotoxic C-20 steroid dendrodoristerol **(3)** (Figure 2) in the Vietnamese nudibranch *Dendrodoris fumata,* highlighting the structural diversity of marine-derived steroids [32]. The structure of dendrodoristerol was elucidated using a combination of advanced spectroscopic techniques, including 1D and 2D NMR, HR-QTOF MS, and circular dichroism (CD). *Nudibranch mollusks*, such as *D. fumata*, are renowned for their unique chemical defense mechanisms, often accumulating bioactive metabolites from their prey [33]. The chemical composition of these mollusks, typically dominated by sesquiterpenes like drimane derivatives, reflects their role in repelling predators. In this study, the focus was on the steroid dendrodoristerol and three known compounds: 5α,6α-epoxy-3β,7α-dihydroxycholest-8(14)-ene, 3β-hydroxycholesta-5,8-dien-7-one, and 5α,8α-epidioxycholest-6-en-3β-ol.

Dendrodoristerol **(3)** is classified as a degraded sterol belonging to the incisterol class, which is characterized by significant structural modifications, including the loss of side-chain carbons and rearrangements in the steroidal nucleus. These modifications are hypothesized to occur through enzymatic oxidation and hydrolysis processes facilitated by marine microbial communities or the host organism itself. Such degradation pathways are common in marine environments, where sterols are subjected to oxidative stress, enzymatic activity, and interactions with symbiotic microorganisms. For instance, similar degradation patterns have been observed in sterols isolated from marine sponges (*Dictyonella incisa*), sea slugs (*Syphonota geographica*), and marine-derived fungi (*Phellinus igniarius*). These processes often result in the formation of highly oxidized or truncated sterols, which exhibit unique biological activities, including cytotoxicity and antimicrobial effects [33,34,35,36]. These sterols have been shown to exhibit various biological activities, including cytotoxic, antimicrobial, and protein tyrosine phosphatase 1B (PTP1B) inhibitory effects. Although *D. fumata* is known to feed on sponges, further studies are needed to determine if dendrodoristerol is synthesized by the mollusk itself or acquired through its diet [32].

In cytotoxic assays, compound **(3)** demonstrated significant activity against six human cancer cell lines (HL-60, KB, LU-1, MCF-7, LNCaP, and HepG2), with IC_50_ values ranging from 21.59 to 41.19 μM [32]. While its cytotoxicity was slightly lower than the positive control ellipticine, its unique structure and bioactivity make it a promising candidate for further exploration. Furthermore, dendrodoristerol **(3)** induced apoptosis in HL-60 cells, as confirmed by morphological changes at concentrations of 15.63 and 31.25 μM with Hoechst 33342 staining. Flow cytometry analysis revealed that at 31.25 μM, dendrodoristerol **(3)** induced early apoptosis (14.11%) and necrosis (5.24%) after 24 h of treatment. Caspase-3 activation further supported its role in triggering apoptosis [32]. The results suggest that dendrodoristerol may be a promising anticancer agent, highlighting the need for further exploration of its therapeutic applications [32].

Five steroids were extracted from the cold-water starfish *Ctenodiscus crispatus*, collected in the Sea of Okhotsk [37]. The steroids were identified using various spectroscopic techniques, including 1D and 2D NMR and mass spectrometry (MS). Among the isolated compounds, (25*S*)-5α-cholestane-3β,5,6β,15α,16β,26-hexaol **(4)** demonstrated significant cytotoxic activity against two human carcinoma cell lines: hepatocellular carcinoma (HepG2) and glioblastoma (U87MG) (Figure 3) [37].

D. J. Xiao et al., from the Guangzhou Institute of Chemistry, reported the discovery of a new steroid from the marine sponge *Cinachyrella australiensis*, collected in the South China Sea (Figure 4) [38]. The isolated compound: (3*E*)-cholest-4-en-3,6-dione-3-oxime **(5)** was characterized using various spectroscopic techniques, including IR, ^1^H NMR, ^13^C NMR, mass spectrometry (MS), DQCOSY, NOESY, TOCSY, HMQC, and HMBC. This steroid was subjected to a bioassay, which demonstrated its cytotoxic properties against Hep G2 liver cancer cells, indicating its potential biological significance [38].

Phallusiasterol C (6), a new disulfated steroid isolated from the Mediterranean tunicate *Phallusia fumigata*, provides insights into the structural diversity and biological roles of marine steroids (Figure 5) [39]. The structure of this compound **(6)** was elucidated through extensive spectroscopic analysis. This steroid features a unique (22*E*)-26,27-dinor-24-methyl-25-hydroxy side chain, making it the first sterol with such a side chain identified from tunicates. This rare structural motif has been observed in only one other sterol, isolated from the starfish *Ctenodiscus crispatus* [40].

The potential role of phallusiasterol C in modulating PXR activity was studied using HepG2 cells (human hepatocarcinoma cell line). Despite its structural similarity to solomonsterol A, a known PXR agonist from the sponge *Theonella swinhoei*, phallusiasterol C did not activate PXR or induce the expression of PXR target genes such as CYP3A4 and MDR1 [39]. This lack of activity highlights the critical structural requirements for PXR modulation, particularly the importance of features in the steroid’s side chain and the configuration around the A/B ring junction. Previous studies have indicated that the sulfate group in the side chain is essential for stabilizing interactions with PXR [41]. However, the failure of phallusiasterol C to exhibit activity suggests that other structural factors, such as the ∆^5^ double bond, may negate the receptor-binding properties observed in related compounds. Although compound **(6)** did not show bioactivity in the PXR assays, this study provides valuable information on the structure–activity relationship of sulfated steroids, particularly in nuclear receptor modulation.

Research on the Vietnamese starfish *Archaster typicus* led to the isolation and characterization of eight highly hydroxylated steroids, including three new compounds **(7–9)** (Figure 6) [42]. These new compounds were identified as sodium salts of (24*S*)-5α-cholestane-3β,4β,5,6α,7β,8,14,15α,24-nonaol 6-sulfate **(7)**, (24*E*)-5α-cholest-24-ene-26-yde-3β,6α,8,14,15α-pentaol 15-sulfate **(8)**, and 5α-cholest-3β,6α,8,14,15α,24,25,26-octaol 15-sulfate **(9)**. The structure elucidation of these compounds was accomplished using advanced spectroscopic techniques, including 1D/2D NMR and Fourier transform ion cyclotron resonance mass spectrometry (FT-ICR-MS). These steroids exhibit unique hydroxylation patterns, which can serve as chemical markers for the taxonomic identification of *A. typicus* [42].

Polyhydroxylated steroids and steroidal glycosides from starfish represent a structurally diverse and biologically active class of marine natural products [43,44]. These compounds typically contain four to nine hydroxyl groups, positioned at strategic locations on the steroidal nucleus and side chains, such as 3β, 6α (or β), 8, 15α (or β), and 16β, contributing significantly to their bioactivity. Asterosaponins, a prominent subclass, are characterized by a 3-O-sulfated 3β,6α-dihydroxysteroid nucleus with a 9(11)-double bond and oligosaccharide chains linked at C-6 [45,46,47,48,49,50,51]. The structural complexity of these compounds is exemplified by pectiniferosides A–J from *Patiria pectinifera*, which feature a pentahydroxycholestane aglycone with sulfated and/or methylated monosaccharides [52].

Recent studies on *Archaster typicus* have identified highly hydroxylated steroids and asterosaponins with unique hydroxylation and sulfation patterns, expanding the known structural diversity of these marine steroids [45,46,47,48,49,50,51]. Comparative analyses with other starfish species, such as *Acanthaster planci* and *Ctenodiscus crispatus*, further highlight the pharmacological relevance of these compounds. Steroidal oligoglycosides from *A. planci* exhibit antimicrobial, anti-inflammatory, and cytotoxic properties, including activity against *Pseudomonas aeruginosa* and MCF-7 breast carcinoma cells [43,44].

Extracts from the starfish *Aphelasterias japonica*, collected off the Russian coast of the Sea of Japan, yielded two new steroids: the disulfated quinovoside aphelasteroside C **(10)** and the monosulfated polyhydroxysteroid aphelaketotriol **(11)** (Figure 7) [53]. Both compounds feature a unique 23-oxo-24-hydroxylated side chain, unprecedented in marine steroids. Additionally, the known compounds cheliferoside L1 (12), 3-O-sulfoasterone **(13)**, forbeside E3 **(14)**, and 3-O-sulfothornasterol A **(15)** were isolated. Biological assays revealed that compounds **(10–12, 14, 15)** exhibited hemolytic activity against mouse erythrocytes, whereas 3-O-sulfoasterone **(4)** was inactive at concentrations below 5 × 10^−4^ M [45]. Structurally, compounds **(10–15)** share key features with asterosaponins, including the 3β,6α-diol moiety, a ∆^9^(^11^) double bond, and a sulfate group at C-3, but they differ in their sugar chains. Unlike asterosaponins, which possess extended five- or six-sugar chains, compounds **(10, 12, 14)** contain only a single sugar residue. The observed differences in steroid composition between Japanese and Russian populations of *A. japonica* may be attributed to ecological factors such as diet and seasonality, or they could indicate taxonomic distinctions [53].

The alcoholic extract of the starfish *Henricia leviuscula*, collected from the Sea of Okhotsk, yielded four previously unreported polyhydroxysteroids **(16–19)** and a novel steroidal glycoside, leviusculoside J **(20)** (Figure 8) [54]. Hemolytic activity assays against mouse erythrocytes demonstrated that compounds **(17)** and **(20)** exhibited membrane-disrupting effects, with HC_50_ values of 2.1 × 10^−4^ M and 8.0 × 10^−5^ M, respectively. At 40 °C, leviusculoside J **(20)** induced complete hemolysis at 8.0 × 10^−5^ M within 5 min, indicating its potent lytic activity [54].

Two new sulfated triterpene glycosides, hemoiedemosides A **(21)** and B **(22)**, were isolated from the Patagonian sea cucumber *Hemoiedema spectabilis* (Figure 9) [55]. Structural elucidation using NMR and FABMS revealed that both compounds share a common aglycon but differ in the sulfation pattern of their tetrasaccharide chains. Hemoiedemoside B **(22)** represents a rare example of a trisulfated triterpene glycoside within the Cucumariidae family. Both glycosides exhibited notable antifungal activity against *Cladosporium cucumerinum*, with hemoiedemoside A **(21)** demonstrating stronger inhibition than the commercial fungicide benomyl at higher concentrations. The presence and number of sulfate groups in the sugar moiety appear to significantly influence bioactivity, as the desulfated derivative 1a showed substantially reduced antifungal potency. Additionally, hemoiedemoside A **(21)** displayed higher brine shrimp toxicity, LC_50_ of 18.7 ppm, compared with hemoiedemoside B **(22)**, which was two times less active [55]. These findings highlight the structural and biological significance of sulfated triterpene glycosides from *Hemoiedema spectabilis*.

Four new compounds, spiculiferosides A **(23)**, B **(24)**, C **(25)**, and D **(26)**, were isolated from the starfish *Henricia leviuscula spiculifera* collected in the Sea of Okhotsk (Figure 10) [56]. Compounds **(23–25)** are monosulfated polyhydroxysteroid glycosides containing two carbohydrate moieties, with one attached to C-3 of the steroid core and another at C-24 of the aglycone’s side chain. Compounds **(24)** and **(25)** are biosides, while compound **(23)** contains three monosaccharide residues, a rare characteristic among polar steroids from starfish. Notably, the presence of a 5-substituted 3-OSO_3_-α-L-Araf unit in these compounds was identified for the first time in starfish-derived steroid glycosides. Cell viability assays revealed negligible cytotoxicity against human embryonic kidney HEK293, melanoma SK-MEL-28, breast cancer MDA-MB-231, and colorectal carcinoma HCT 116 cells at concentrations up to 100 μM [56]. However, compounds **(23–25)** significantly inhibited the proliferation and colony formation of HCT 116 cells, with compound **(25)** showing the most potent effects. Compound **(25)** induced dose-dependent G2/M cell cycle arrest and regulated the expression of cell cycle proteins (CDK2, CDK4, cyclin D1, p21), while also inhibiting the phosphorylation of c-Raf, MEK1/2, and ERK1/2 kinases of the MAPK/ERK pathway. The effects of compounds **(23–25)** on cell proliferation were further assessed, with compound **(25)** exhibiting the greatest antiproliferative activity. Compound **(25)** at 100 µM suppressed HCT 116 cell proliferation by 55%, 57%, and 60% after 24, 48, and 72 h, respectively. In soft agar colony formation assays, compounds **(23–25)** reduced colony numbers of HCT 116 cells in a dose-dependent manner, with compound **(25)** showing the most significant inhibition [56]. These findings highlight compound **(25)** as a promising candidate for further studies on its molecular mechanisms in colorectal cancer treatment. Further analysis of the compound **(25)** mechanism revealed that its anticancer activity involves the regulation of cell cycle proteins and inhibition of the ERK1/2 MAPK signaling pathway, suggesting that compound antiproliferative effects are likely mediated through these pathways [56].

From the ethanolic extract of the Japanese slime sea star *Pteraster marsippus*, two new steroid 3β,21-disulfates **(27, 28)** and two new steroid 3β,22- and 3α,22-disulfates **(29, 30)** were isolated (Figure 11) [57]. The structures of compounds **(27–30)** were determined using detailed one-dimensional and two-dimensional NMR, HR ESI MS, and HR ESI MS/MS data. Compounds **(27)** and **(28)** possess a Δ22-21-sulfoxy-24-norcholestane side chain, while compounds **(29)** and **(30)** contain a Δ24(28)-22-sulfoxy-24-methylcholestane side chain, a feature first discovered in the polar steroids of starfish and ophiuroids. The cytotoxic effects of compounds **(27–30)** were evaluated using the human breast cancer cell lines T-47D, MCF-7, and MDA-MB-231 [57]. Compounds **(27)** and **(28)** exhibited significant colony-inhibiting activity against T-47D cells, while compounds **(29)** and **(30)** demonstrated stronger effects on MDA-MB-231 cells. In cell viability assays, compounds **(27–30)** inhibited cell viability at 50 µM in a dose-dependent manner, with inhibition rates ranging from 5% to 22% across the different cell lines, lower than that of cisplatin, which was used as a positive control. Compound **(30)** showed the highest inhibition of MDA-MB-231 cells. In colony formation assays, compounds **(27–30)** inhibited colony formation significantly in a dose-dependent manner. Compound **(27)** inhibited colony formation by up to 76% in T-47D cells, compound **(28)** by up to 86% in T-47D cells, compound **(29)** by up to 87% in MCF-7 cells, and compound **(30)** by up to 90% in MDA-MB-231 cells at the highest tested concentration of 50 µM [57]. These results indicate that compounds **(27)** and **(28)** are most effective against T-47D cells, while compounds **(29)** and **(30)** exhibit stronger effects on MDA-MB-231 cells.

Four new disulfated steroids **(31–34)**, including three 3β,21-disulfates **(31–33)** and one 3β,22-disulfate **(34)**, were isolated from the ethanolic extract of *Pteraster marsippus*, a Far Eastern starfish species (Figure 12) [58]. The structures of these compounds were determined using detailed NMR and HR ESI MS analyses. Steroids **(31)** and **(32)** contain an oxo group at position C-7 in the steroid nucleus and differ by the presence of the conjugated 5(6)-double bond. Compound **(34)** features a novel Δ24-22-sulfoxycholestane side chain not previously described in starfish or ophiuroid steroids. The cytotoxic activities of compounds **(31–34)** were evaluated in 2D and 3D cultures of human epithelial kidney cells (HEK293), melanoma cells (SK-MEL-28), small intestine carcinoma cells (HuTu80), and breast carcinoma cells (ZR-75-1) using the MTS assay. The mixture of **(31)** and **(32)** showed significant inhibitory effects on the viability of ZR-75-1 breast carcinoma cells, with IC_50_ values of 90.4 µM in 2D and 21.9 µM in 3D cultures, indicating their potential as antineoplastic agents [58]. Other tested compounds exhibited less pronounced effects [58]. These findings suggest that the combination of **(31)** and **(32)** has the highest cytotoxicity among the compounds studied, and their structural features, including the oxo group and conjugated double bond, play a critical role in their biological activity.

Four new conjugates, esters of poly(hydroxy)steroids with long-chain fatty acids, were isolated from the deep-sea starfish *Ceramaster patagonicus* (Figure 13) [59]. After detailed structural analysis using NMR techniques, ESI MS, and chemical transformations, the authors determined that compounds **(35–38)** share a common steroidal core of 5α-cholestan-3β,6β,15α,16β,26-penta(hydroxy)steroid and differ by the following fatty acid units: 5′Z,11′Z-octadecadienoic **(35)**, 11′Z-octadecenoic **(36)**, 5′Z,11′Z-eicosadienoic **(37)**, and 7′Z-eicosenoic **(38)** acids. Compounds **(35–38)** exhibit cytotoxic activity, though their effects vary depending on the cell type. To evaluate the cytotoxic effects of these compounds, several human and mouse cell lines were tested. The JB6 Cl41 cell line, derived from mouse epidermis, was used to assess potential effects on skin cancer. This cell line is sensitive to tumor promoters, which makes it a relevant model for studying the early stages of skin cancer. Compounds **(35)**, **(36)**, and **(37)** caused 50% growth inhibition of JB6 Cl41 cells at concentrations of 81, 40, and 79 µM, respectively [59]. These results suggest that these compounds may modulate the pathways involved in skin cancer development. The MDA-MB-231 cell line, a human triple-negative breast cancer model, was also tested. This subtype of breast cancer is aggressive and resistant to conventional hormonal therapies. Compounds **(35)**, **(36)**, and **(37)** showed IC_50_ values of 74, 33, and 73 µM, respectively, indicating significant cytotoxicity in this cell line. Notably, compound **(36)** was particularly effective at 20 µM, significantly inhibiting colony formation and migration of MDA-MB-231 cells. The migration of MDA-MB-231 cells was reduced by 50%, while compound **(36)** completely inhibited the migration of HCT 116 cells, a human colon carcinoma line, at the same concentration. HCT 116 cells, known for their ability to form tumors in immunodeficient mice, were also tested. The IC_50_ values for compounds **(35)**, **(36)**, and **(37)** were 73, 31, and 71 µM, respectively. Furthermore, migration assays revealed that compound **(36)** completely halted the migration of HCT 116 cells at 20 µM. In contrast, compounds **(35)**, **(37)**, and **(38)** showed more moderate effects in limiting HCT 116 cell migration [59]. It is important to note that the compounds did not exhibit selective toxicity toward cancer cells alone. Normal cells were also affected, albeit to a lesser extent. To further investigate their effects on cancer-related processes, the authors performed additional studies using soft agar and wound healing assays. The studies suggest that polyhydroxylated steroids such as those isolated from starfish may play an important role in transporting fatty acids to peripheral tissues, similar to cholesterol in vertebrates. These steroids are found predominantly in the digestive organs of starfish, with the highest concentrations located in the stomach and pyloric ceca, suggesting their biological importance in marine organisms [59].

The new sterol (23R)-methoxycholest-5,24-dien-3β-ol **(39)** was isolated from the marine bryozoan *Cryptosula pallasiana* collected at Huang Island, China (Figure 14) [60]. This sterol features a rare methoxy group at C-23 and a double bond between C-24 and C-25, setting it apart from the typical sterols found in marine organisms. Its structure was determined through extensive spectroscopic analysis, including quantum electronic circular dichroism (ECD) calculations. In terms of biological activity, compound **(39)** was evaluated for cytotoxicity against the human tumor cell lines HL-60 (human myeloid leukemia), HepG2 (human hepatocellular carcinoma), and SGC-7901 (human gastric carcinoma). The results indicated that (23R)-methoxycholest-5,24-dien-3β-ol **(39)** displayed moderate cytotoxicity, with IC_50_ values ranging between 12.34 μM and 18.37 μM across the tested cell lines [60].

### 2.2. Soft Corals

Research on *Dichotella gemmacea* led to the isolation of one new 19-oxygenated steroid, 25-acetate-nebrosteroid K **(40)**, and five known steroids, along with one known 19-hydroxy steroidal glycoside—Junceelloside C (Figure 15) [61]. The new compound **(40)** is significant as the first steroid with a 19-oic acid methyl ester group isolated from gorgonians, representing a novel structural feature within this class of compounds.

The structure and relative configuration of 25-acetate-nebrosteroid K **(40)** were elucidated using various spectroscopic techniques, including NOESY, along with 1D and 2D NMR methods. Compound **(40)** belongs to the class of 19-oxygenated steroids, which are rarely found in gorgonians. Its methyl ester group at C-19 serves as a key structural feature, distinguishing it from other known gorgonian steroids. The biological activity of the isolated steroids was evaluated for lethal activity against *Artemia salina* (brine shrimp) and cytotoxicity against A549 and HL-60 cell lines [61]. Compound **(40)** exhibited strong lethality toward *A. salina*, with a lethality rate of 75.9% at 25 μg/mL, indicating its potential as a bioactive compound. However, it did not demonstrate significant cytotoxic activity [61].

Six new steroids—Klyflaccisteroids **(41–46)**—were isolated from the soft coral *Klyxum flaccidum* (Figure 16) [62]. Compound **(46)** is notable as the first 9,11-secogorgosteroid 11-carboxylic acid isolated from natural sources. The cytotoxicity and anti-inflammatory activities of these compounds were evaluated, revealing promising bioactivity in several cases. Compound **(41)** selectively inhibited the growth of human lung adenocarcinoma (A549) cells with an ED_50_ of 7.7 μg/mL. Compound **(46)** exhibited the strongest cytotoxicity toward human colon carcinoma (HT-29) cells, with an ED_50_ of 6.9 μg/mL [62]. Additionally, compounds **(43)** and **(45)** demonstrated potent cytotoxic effects against A549 and murine leukemia (P388) cells, with ED_50_ values of 6.1 and 3.7 μg/mL, respectively [62].

Anti-inflammatory assays showed that compounds **(43)** and **(46)** exhibited strong inhibition of superoxide anion generation and elastase release in human neutrophils, with inhibition percentages exceeding 75% at 10 μM. Most compounds displayed cytotoxicity against specific cancer cell lines, while compounds **(43)** and **(46)** exhibited significant anti-inflammatory effects [62].

Twelve steroids, including five new compounds **(47–51)**, were isolated from the methanol extract of the Vietnamese soft coral *Sinularia conferta* (Figure 17) [63]. The investigation focused on their cytotoxic effects against three human cancer cell lines: lung carcinoma (A-549), cervical adenocarcinoma (HeLa), and pancreatic epithelioid carcinoma (PANC-1). Evaluation using MTT assays revealed that one of the new compounds, ergosta-24(28)-ene-3β,5α,6β-triol-6-acetate **(52)**, exhibited potent cytotoxic activity against all three tested cell lines. The IC_50_ values were 3.64 ± 0.18 µM for A-549, 19.34 ± 0.42 µM for HeLa, and 1.78 ± 0.69 µM for PANC-1 [63]. Compound **(52)** was notably more effective than standard anticancer agents such as camptothecin and etoposide in A-549 and HeLa cells. It also displayed comparable potency to etoposide in PANC-1 cells. Structural analysis suggested that the 3β,5α,6β-triol motif, along with an acetate group at the 6-position, may play a key role in the observed cytotoxicity [63].

The presence of the acetate group at C-6 appears to enhance the compound’s lipophilicity, potentially improving its interaction with cellular targets and membrane permeability. This structural feature likely contributes to the significantly higher cytotoxic activity of compound **(52)** compared with its non-acetylated analog, ergosta-24(28)-ene-3β,5α,6β-triol **(51)**, which exhibited markedly lower activity (IC_50_ > 100 µM). These findings highlight the critical role of the acetate group in modulating the biological activity of steroids [63].

This study expands the chemical profile of *Sinularia conferta*, a relatively understudied species, and identifies promising bioactive compounds for potential therapeutic applications in cancer treatment [63]. The results underscore the importance of structural modifications, such as acetylation, in enhancing the pharmacological properties of marine-derived steroids.

Researchers reported the discovery of two new steroid derivatives: erectsterate A **(53)** and erectsterate B **(54)** (Figure 18) [64]. These compounds, epimers at the C-10 position, were isolated from the soft coral *Sinularia erecta,* collected in the South China Sea. Their structures are characterized by significant degradation of the B ring and an ester bond linking rings A and C/D. Structures were confirmed using advanced spectroscopic techniques. Interestingly, these compounds share structural similarities with the previously identified Chaxin B **(55)** [65]. However, this discovery marks the first report of such steroids in soft corals. A novel biosynthetic pathway, distinct from that of chaxins, has also been proposed. Understanding the biosynthetic pathways of natural products is crucial for elucidating their origin, structural diversity, and potential for drug discovery. In the context of marine-derived steroids, these pathways often involve complex enzymatic processes that lead to unique structural features. During the biosynthesis of erectsterates A and B, Baeyer–Villiger oxidation occurred in the C ring, resulting in the formation of a unique seven-membered lactone ring—an unprecedented feature in steroid chemistry [64]. The discovery of this pathway not only explains the biosynthesis of these compounds but also provides insights into potential synthetic strategies for creating structurally similar bioactive molecules.

Preliminary biological activity assays revealed that erectsterate B **(54)** exhibited weak cytotoxicity against four cancer cell lines: A549, HT-29, SNU-398, and Capan-1. These results suggest potential avenues for further research into its application in anticancer therapies, although more extensive studies are needed to fully assess its therapeutic efficacy [64,65].

Hurgadacin **(56)** is a newly discovered steroid isolated from the soft coral *Sinularia polydactyla*, collected from the Red Sea near Hurghada (Figure 19) [66]. The newly identified steroid compound **(56)** was found alongside other closely related compounds, 24-methylenecholestane-3β,5α,6β-triol **(57)** and 24-methylenecholestane-1α,3β,5α,6β,11α-pentol **(58)**. The biological activity of the extracts and isolated components was assessed, focusing on their antibacterial activity against various pathogenic microorganisms and cytotoxicity, examined with a shrimp mortality test [66]. The biological tests performed highlight the potential of these compounds. Corals from the Red Sea are known for their unique chemistry, which may provide valuable compounds for research into antibiotic resistance and antineoplastic therapy [67,68,69].

However, the isolated compounds, including the new steroid, were found to be inactive against *Bacillus subtilis*, *Staphylococcus aureus*, *Streptomyces viridochromogenes* (Tü57), *Escherichia coli*, *Candida albicans*, *Mucor miehi*, *Chlorella vulgaris*, *Chlorella sorokiniana*, *Scenedesmus subspicatus*, *Rhizoctonia solani, and Pythium ultimum*. In contrast, the crude coral extract displayed significant activity against the phytopathogenic fungus R. solani [66].

The Red Sea soft coral *Dendronephthya* sp. has yielded a novel cytotoxic steroid, dendronestadione **(59)**, featuring a C-27 carbocyclic skeleton with two α,β-unsaturated carbonyl groups (Figure 20) [70]. This structural motif enhances cytotoxic activity, particularly against prostate adenocarcinoma (PC-3) cells, with an IC_50_ value of 7.8 ± 0.80 µM. Additionally, dendronestadione **(59)** exhibited notable antiproliferative effects against hepatocellular carcinoma (HepG2) and colorectal adenocarcinoma (HT-29) cells, with IC_50_ values of 19.1 ± 1.81 µM and 32.4 ± 2.84 µM, respectively [70]. The pronounced cytotoxicity observed for this ketosteroid suggests a structure–activity relationship wherein the presence of α,β-unsaturated carbonyl functionalities contributes to enhanced biological activity.

The novel steroid 3β,7β,9α-trihydroxycholest-5-en **(60)** was isolated from *Petrosia* sp. (Figure 21) [71]. This polyoxygenated steroid features a unique arrangement of hydroxyl groups at the 3, 7, and 9 positions and a conjugated double bond at the 5 position, characteristic for sterol compounds. The chemical structure of compound **(60)** was elucidated through detailed 1D and 2D NMR, UV, IR, and MS spectroscopy. Its cytotoxicity was evaluated against the HepG2 and MCF-7 cancer cell lines, with an IC_50_ value in the range of 20 to 500 μM [71], suggesting its potential as a bioactive compound. Furthermore, compound **(60)** demonstrated a significant affinity for DNA, which may underlie its cytotoxic effects.

Three marine steryl hexadecanoates, 3β-hexadecanoylcholest-5-en-7-one **(61)**, 3β-hexadecanoylcholest-5-en-7β-ol **(62)**, and cholest-5-en-3β-yl-formate **(63)**, were isolated from the anthozoan black coral *Antipathes dichotoma* (Figure 22) [72]. The structures of these compounds were determined through extensive 1D (^1^H, ^13^C, and DEPT) and 2D (COSY, HSQC, and HMBC) NMR, UV, IR, and MS spectroscopic analyses. Compound **(61)** features a 7-one functional group and a hexadecanoyl moiety at the 3β position, while compound **(62)** is similar but contains a hydroxyl group at position 7β. Compound **(63)**, a known steryl derivative, consists of a formate ester group at the 3β position. In terms of biological activity, compounds **(61)** and **(63)** exhibited moderate cytotoxic effects, with IC_50_ values ranging from 28.7 to 48.2 μg/mL across several cancer cell lines, including HepG2, WI 38, VERO, and MCF-7 [72]. Compound **(62)** showed weaker cytotoxic activity, with IC_50_ values ranging from 70.5 to 84.8 μg/mL [72].

From the black coral *Antipathes dichotoma*, a new steroid, (22E)-methylcholesta-5,22-diene-1α,3β,7α-triol **(64)**, along with three known steroid derivatives, 3β,7α-dihydroxy-cholest-5-ene **(65)**, (22E,24S),5α,8α-epidioxy-24-methylcholesta-6,22-dien-3β-ol **(66)**, and (22E,24S),5α,8α-epidioxy-24-methylcholesta-6,9(11),22-trien-3β-ol **(67)**, were isolated (Figure 23) [73]. The structures of these steroids were determined using NMR spectroscopic analysis and comparisons with data from the literature. Among the steroids tested for antibacterial activity, compound **(64)**, the trihydroxy steroid, exhibited potent activity against *Bacillus subtilis* and *Pseudomonas aeruginosa*. The epidioxy steroids **(66)** and **(67)** showed similar strong antibacterial activity, particularly against *B. subtilis* [73].

From the Red Sea sponge *Echinoclathria gibbosa*, two new steroid compounds, β-sitosterol-3-O-(3Z)-pentacosenoate **(68)** and 5α-pregna-3β-acetoxy-12β,16β-diol-20-one **(69)**, were isolated (Figure 24) [74]. The structures of these compounds were determined through 1D and 2D NMR spectroscopy and mass spectrometry. Compound **(68)** is a steryl ester derivative, which combines the well-known sterol β-sitosterol with a long-chain fatty acid, 3Z-pentacosenoic acid. Compound **(69)** is a pregnane-type steroid with acetoxy, diol, and ketone functionalities at positions 3, 12, and 20. These compounds and the total MeOH extract and fractions of *E. gibbosa* were screened for antimicrobial, anti-inflammatory, antipyretic, and hepato-protective activities. Compound **(68)** exhibited weak growth inhibitory activity against three human cancer cell lines—A549 (non-small-cell lung cancer), U373 (glioblastoma), and PC-3 (prostate cancer). Compound **(69)** was found to be inactive in the same assays [74].

### 2.3. Marine and Mangrove-Associated Fungi

Marine-derived fungi, particularly those associated with algae, are an emerging source of bioactive secondary metabolites [75,76,77]. Despite being relatively understudied, these fungi hold potential for novel pharmacological applications.

Research on the deep-sea-derived fungus *Rhizopus* sp. W23 led to the identification of six new cyclocitrinol analogs **(70–72, 75, 77–78)** and twelve known compounds **(73–74, 79–88)** (Figure 25) [78]. These compounds feature a unique 7/7/6/5-tetracyclic scaffold with bicyclo [4.4.1] A/B rings. Notably, norcyclocitrinoic acids A **(70)** and B **(71)** represent only the second instance of 24,25-bisnor cyclocitrinols in natural products. The structures of these new steroids were elucidated through detailed spectroscopic analysis and X-ray crystallography [78]. The stereochemistry at C-24 remains undefined due to the lack of data.

Cyclocitrinol analogs are an uncommon class of natural products, first discovered from the terrestrial fungus *Penicillium citrinum* [78]. This subgroup of steroids is characterized by minor modifications of the tetracyclic scaffold, particularly in the A/B ring system. The key modifications include variations in the degree of unsaturation, especially the presence of a double bond, and alterations in substituent patterns These modifications may include changes in the hydroxylation and methylation patterns in the tetracyclic structure, which are crucial for the biological activity of these compounds. These compounds have shown a variety of biological properties such as antibacterial properties and moderate cytotoxicity against cancer cell lines. The structural diversity within this group increases their scientific significance [78].

Compound **(82)** is particularly noteworthy for its ability to enhance osteoblastogenesis while inhibiting adipogenesis in mature bone marrow stromal cells at 5 μM [78]. It suggests a potential as an anti-osteoporosis agent. Osteoporosis is a major clinical challenge marked by diminished bone density. It is commonly treated through therapies that either inhibit osteoclast activity or promote osteoblast function. Among the tested compounds, **(82)** and **(87)** were the most effective, showing 78% and 65% inhibition of adipogenic differentiation in BMSCs, respectively, alongside enhanced mineralization of osteoblasts. Importantly, these compounds exhibited no significant cytotoxicity (IC_50_ > 20 μM), further highlighting their potential as safe therapeutic agents [78]. This research points to the promising potential of cyclocitrinol analogs as candidates for osteoporosis treatment, offering a rare example of natural products with osteogenic activity.

Qiao et al. detail the discovery of Asporyergosterol **(89)**, a novel steroid characterized by an E double bond between C-17 and C-20 (Figure 26) [79]. This compound was isolated from the endophytic fungus *Aspergillus oryzae*, which was found in the marine red alga *Heterosiphonia japonica*. Alongside asporyergosterol, four known steroids were also identified: (22*E*,24*R*)-ergosta-4,6,8(14),22-tetraen-3-one, (22*E*,24*R*)-3β-hydroxyergosta-5,8,22-trien-7-one, (22*E*,24*R*)-ergosta-7,22-dien-3β,5α,6β-triol, and (22*E*,24*R*)-5α,8α-epidioxyergosta-6,22-dien-3β-ol. The authors suggest that all isolated steroids from *Aspergillus oryzae* showed low activity in modulating acetylcholinesterase [79].

From the fermentation broth of the endophytic fungus *Penicillium oxalicum* HLLG-13, isolated from the roots of the mangrove tree *Lumnitzera littorea*, fifteen secondary metabolites were identified, including the newly discovered steroid andrastin H **(90)** (Figure 27) [80]. The structure of this compound was elucidated using NMR and HR ESI MS spectral analysis, and its absolute configuration was confirmed through quantum chemical electronic circular dichroism (ECD) calculations. Compound **(90)** exhibited potent antibacterial activity, particularly against *Staphylococcus epidermidis* and *Candida albicans*, with minimum inhibitory concentration (MIC) values ranging from 6.25 to 25 μg/mL [80]. The presence of a steroidal framework in this compound suggests a potential mode of action involving membrane disruption or inhibition of key enzymatic pathways in microbial cells.

In addition to its antimicrobial properties, andrastin H **(90)** demonstrated significant insecticidal activity against newly hatched larvae of *Helicoverpa armigera*, an important agricultural pest. The compound exhibited an IC_50_ value of 50 μg/mL, comparable to the widely used botanical insecticide azadirachtin [80]. This bioactivity highlights the ecological role of fungal steroids as chemical defense agents and underscores their potential for agricultural applications. The discovery of andrastin H **(90)** adds to the structural diversity of bioactive steroids from fungal sources and reinforces the importance of marine-derived fungi as a reservoir of novel steroidal scaffolds with promising pharmacological and agrochemical applications.

The chemical investigation of a mangrove-derived *Aspergillus* sp. led to the identification of two structurally unique steroids, ergosterdiacids A **(91)** and B **(92)**, featuring an unusual 6/6/6/6/5 pentacyclic system (Figure 28) [81]. These compounds were characterized using comprehensive spectroscopic techniques, including 1D and 2D NMR, HR ESI MS, and quantum chemical ECD calculations. Their biosynthetic origin suggests a rare natural Diels–Alder addition. Ergosterdiacids A **(91)** and B **(92)** demonstrated notable biological activity, particularly in the inhibition of the *Mycobacterium tuberculosis* protein tyrosine phosphatase B (MptpB), a validated target for tuberculosis treatment, with IC_50_ values of 15.1 and 30.1 μM, respectively [81]. Molecular docking analyses suggested that their carboxyl groups play a critical role in binding to key active site residues, similar to the known inhibitor OMTS. This inhibition of MptpB highlights the potential of these steroids as lead compounds for drug development. Beyond their antitubercular properties, both compounds exhibited strong anti-inflammatory activity by suppressing nitric oxide (NO) production in LPS-induced RAW 264.7 macrophages. At concentrations of 4.5 and 3.6 μM, respectively, ergosterdiacids A **(91)** and B **(92)** significantly reduced NO levels without cytotoxic effects. However, at concentrations exceeding 10 μM, partial suppression of macrophage viability was observed, indicating potential cytotoxicity at higher doses [81].

The coral-associated fungus *Penicillium oxalicum* HL-44, isolated from *Sinularia gaweli*, produced a new ergostane-type sterol ester **(93)** alongside several structurally related derivatives (Figure 29) [10]. The structure of **(93)** was determined using NMR, HR-MS, and other spectroscopic techniques, revealing a rare modification within the ergosteroid framework. Notably, this is the first report of such a sterol ester from the *Eurotiaceae* family, highlighting its unique biosynthetic origin. Biological evaluation of **(93)** demonstrated significant anti-inflammatory activity in *Raw264.7* macrophage cells [10]. At a concentration of 20 μM, **(92)** reduced *Ifnb1* expression by 67.4%, suggesting strong suppression of the cGAS-STING pathway. Additionally, sterols **(94)** and **(95)** exhibited complementary inhibitory effects on *Tnfα* expression, with reductions of 52.3% and 48.7%, respectively, at the same concentration [10].

Solitumergosterol A **(96)**, a unique C30 steroid, was isolated from the deep-sea-derived fungus *Penicillium solitum* MCCC 3A00215 (Figure 30) [11]. Its structure, featuring a 6/6/6/6/5 pentacyclic carbon skeleton, was determined using 1D and 2D NMR, HR ESI MS, and ECD spectra. Compound **(96)** is a Diels–Alder adduct formed through the cyclization of a steroid with an alien motif, likely maleic acid or maleimide. It exhibited weak in vitro antiproliferative activity against MB231 cells, with an inhibition rate of 44.1% at 20 μM, via an RXRα-dependent mechanism [11]. Compound **(96)** attenuates the transcriptional activity of RXRα, as measured using a luciferase reporter assay system. In this experimental setup, the luciferase gene is placed under the control of RXRα-responsive elements, allowing for the quantitative measurement of RXRα-mediated transcription. The results show a relative luciferase activity of 68.6% at 50 μM, which parallels the activity of the positive control, 9-cis-retinoic acid, at 0.1 μM. This decrease in luminescence directly reflects reduced RXRα-mediated gene transcription, as the light output in this system is proportional to the transcriptional activity of RXRα. When RXRα is activated, it binds to its responsive elements and promotes the transcription of the luciferase gene, resulting in increased luminescence. Conversely, when RXRα activity is attenuated, as observed with compound **(96)**, there is a decrease in luciferase gene transcription, leading to reduced luminescence. This suggests that solitumergosterol A **(96)** might influence cancer cell proliferation through the retinoid X receptor (RXR) pathway. The observed reduction in RXRα transcriptional activity implies that compound **(96)** may act as a partial antagonist or modulator of RXRα, which is significant given RXRα’s known roles in various cellular processes, including cell growth, differentiation, and apoptosis. The proposed biosynthetic pathway, which involves a Diels–Alder cycloaddition followed by decarboxylation and aromatization, adds new insights into steroid chemistry, inspiring further research into its chemical and biological synthesis [11]. This unique biosynthetic pathway, coupled with the compound’s ability to modulate RXRα activity, opens up possibilities for developing new therapeutic strategies targeting the RXR pathway in cancer treatment.

## 3. Terrestrial Organisms

### 3.1. Plants

The article “Chenisterol, a New Antimicrobial Steroid from *Chenopodium badachschanicum*” presents the first phytochemical investigation of *Chenopodium badachschanicum*, a member of the *Chenopodiaceae* family [82]. Several plants from this genus are widely used in traditional medicine. Their secondary metabolites display various pharmacological properties, including antimicrobial, antiviral, antifungal, anthelmintic, antioxidant, trypanocidal, antineoplastic, and immunomodulatory activities [83]. This study led to the isolation of a new steroid named chenisterol **(97)** (Figure 31). Known compounds such as ergosterol, stigmasterol, and fucosterol were also isolated. The structure of chenisterol was determined using spectroscopic techniques, including HR ESI-MS along with 1D and 2D NMR [82].

Chenisterol exhibited significant antimicrobial activity against three Gram-positive bacterial strains: *Staphylococcus epidermidis*, *Bacillus subtilis,* and *Corynebacterium diphtheriae* [82]. It also showed activity against one Gram-negative strain, *Klebsiella pneumoniae*. These results suggest that chenisterol has potential as a novel antimicrobial agent and warrants further exploration for therapeutic applications.

The chemical investigation of compounds isolated from the roots of *Codonopsis pilosula* var. *modesta* identified a total of 15 compounds, including five steroids: α-spinasterol **(99)**, β-sitosterol **(100)**, stigmasterol, β-sitosterol-3-O-glucoside, and stigmasterol-3-O-glucoside [84]. Additionally, one new tirucallane-type triterpenoid, codopitirol A **(98)**, was isolated and identified (Figure 32). The structures of these steroids and codopitirol A **(98)** were confirmed using spectroscopic techniques, including HR ESI-MS and 1D/2D NMR [84].

These steroids were evaluated for their α-glucosidase inhibitory activity. Notably, α-spinasterol **(99)** and β-sitosterol **(100)** exhibited promising potential as antidiabetic agents [84]. This study emphasizes the importance of steroids in the pharmacological activities of *Codonopsis pilosula*, particularly regarding their potential role as inhibitors of enzymes related to diabetes management. It also highlights the significance of these findings, especially considering that only a few phytochemical investigations have been conducted on *C. pilosula* var. *modesta*.

The discovery of turpesteryl ester **(101)**, a novel steroid isolated from *Ipomoea turpethum* (L.) R.Br., enhances the understanding of bioactive compounds from medicinal plants (Figure 33) [85]. The structure of turpesteryl ester was confirmed through spectroscopic techniques, including 2D NMR. *Ipomoea turpethum*, native to various parts of Asia, is used in traditional Unani and Ayurvedic medicine. It belongs to the *Convolvulaceae* family, which includes about 40 genera and 1200 species of climbing or twining herbaceous plants [86]. The genus *Ipomoea* is well known for its wide distribution in tropical regions. Despite its medicinal relevance, little research has been conducted on its chemical constituents [87]. Previous studies on this species reported only an acrylamide derivative, making this study the first to isolate and characterize turpesteryl ester and other associated compounds [85]. Turpesteryl ester **(101)** demonstrated significant antibacterial activity against both Gram-positive (*Staphylococcus aureus* and *Micrococcus luteus*) and Gram-negative (*Escherichia coli*, *Klebsiella pneumoniae*, and *Pseudomonas aeruginosa*) bacterial strains, as shown by the agar well diffusion method [85]. This bioactivity suggests its potential as a natural antibacterial agent. Additionally, positive Salkowski and Lieberman–Burchard tests confirmed the steroidal nature of the compound, while its brisk reaction with dilute NaHCO_3_ indicated the presence of a free carboxylic acid group, which may contribute to its biological activity [85].

Research on *Cyperus rotundus* L. led to the identification of a new steroid glycoside—sitosteryl (6′-hentriacontanoyl)-α-D-galactopyranoside **(102)** (Figure 34) [88]. The structure of this steroid glycoside was determined through 1D and 2D NMR spectroscopic analysis. This analysis revealed it as an ester of sitosterol glycoside attached to a fatty acid. The biological activity of sitosteryl (6′-hentriacontanoyl)-α-D-galactopyranoside **(102)** was notable, demonstrating strong cytotoxicity against L5178y mouse lymphoma cells. It also showed activity in the brine shrimp lethality test, suggesting its potential for further biomedical studies [88].

Investigation of the biologically active compounds from *Fritillaria thunbergii* led to the isolation of steroidal alkaloids **(103–107)** (Figure 35), including two new compounds, Frithunbol A and B **(103–104)** [89]. The structures of these compounds were determined using 1D and 2D NMR spectroscopy along with HR-FAB-MS techniques. The compounds were evaluated for their anti-neuroinflammatory and neuroprotective effects in vitro [89].

The anti-neuroinflammatory effects were tested by measuring nitric oxide (NO) levels in LPS-stimulated BV-2 microglial cells. Compounds **(103–104)** and **(107)** exhibited significant NO inhibition, with IC_50_ values of 16.35, 11.45, and 18.02 μM, respectively. These values surpassed the NO inhibition of the positive control, NG-monomethyl-L-arginine (L-NMMA), which had an IC_50_ value of 19.45 μM [89]. Notably, these compounds were non-cytotoxic at a concentration of 20 μM, indicating selective anti-inflammatory action without harming the cells. The neuroprotective potential of the isolated compounds was assessed by measuring nerve growth factor (NGF) secretion in C6 glioma cells. Compound **(106)** significantly stimulated NGF release, enhancing levels by 134.81% without cytotoxic effects, comparable to the positive control, 6-shogaol, which stimulated NGF release by 134.08% [89]. These findings suggest that these compounds are potential candidates for further studies on their therapeutic applications in neuroinflammation and neurodegeneration.

Two steroid alkaloids, tomatidine **(108)** and solasodine **(109)**, were isolated from the berries of *Solanum aculeastrum*, a medicinal plant traditionally used in South Africa for cancer treatment (Figure 36) [90]. These compounds were evaluated for their anticancer activity against HeLa, MCF7, and HT29 cancer cell lines. The IC_50_ values indicated that both tomatidine and solasodine had the strongest inhibitory effect on HeLa cells. A synergistic effect was observed when the compounds were combined. Cell cycle analysis revealed that tomatidine and solasodine blocked the cell cycle in the G0/G1 phase after 24 h of exposure, increasing the percentage of cells in this phase. However, they showed low apoptotic activity. While both compounds did not induce significant apoptosis, they increased necrotic cell death, particularly at higher concentrations [90].

Researchers investigated the antidiabetic potential of the methanolic extract from the male flowers of *Borassus flabellifer* (*Palmae*) (Figure 37) [91]. This plant, widely distributed and cultivated in tropical Asian countries, has been traditionally utilized for its medicinal properties. The fruit pulp and sap of *B. flabellifer* are commonly used in traditional dishes and as a sweetener for diabetic patients. The study found that the methanolic extract of the male flowers significantly inhibited the increase in serum glucose levels in sucrose-loaded rats when administered orally at a dose of 250 mg/kg. Six new spirostane-type steroid saponins, named borassosides A–F **(110–115)**, were isolated from this extract. The structures of borassosides A–F were elucidated through chemical and physicochemical analyses [91].

Additionally, the principal steroid saponin, Dioscin **(116)**, was tested and demonstrated an inhibitory effect on serum glucose levels in sucrose-loaded rats at a dose of 50 mg/kg orally [91]. This effect was comparable to or greater than that of metformin hydrochloride, which is recognized for its action on intestinal glucose absorption and peripheral insulin sensitivity. Dioscin’s effect surpassed that of metformin hydrochloride in this model. While previous research has identified various steroidal saponins and other constituents from *B. flabellifer*, this study marks the first detailed characterization of the flower parts of the plant [92,93].

Researchers reported the discovery of a new steroid, albosteroid **(117)** (Figure 38) [94]. It was isolated from the *Morus alba*, a plant known for its medicinal properties. This plant has been traditionally used in Asian countries for anti-inflammatory, hypoglycemic, hypolipidemic and antioxidant purposes [95,96,97,98,99,100,101,102]. Compounds were isolated using column chromatography, and their molecular structures were determined through IR, UV, ^1^H and ^13^C NMR, and mass spectroscopic analysis [94]. Among the isolated compounds, albosteroid demonstrated significant anti-ulcer activity in two models: pylorus-ligation-induced and ethanol-induced ulcers. In both models, compound **(117)** substantially reduced the number and severity of ulcers in the stomach [94].

In animal models, albosteroid **(117)** not only decreased ulcer formation but improved other biochemical markers associated with gastric damage [94]. In the ethanol-induced ulcer model, where alcohol triggers oxidative stress and inflammation, albosteroid **(117)** exhibited dose-dependent protective effects. It enhanced the activity of key antioxidant enzymes, such as superoxide dismutase, catalase, and glutathione peroxidase, which help neutralize oxidative damage. Compound **(117)** increased levels of reduced glutathione while lowering elevated levels of glutathione reductase and lipid peroxidation, both markers of oxidative stress. The protective effect on the gastric lining was further confirmed by histopathological analysis of stomach tissues from treated animals, which showed significantly less damage in the groups administered albosteroid [94]. The findings indicate the compound’s potential not only to prevent ulcer formation but also to reverse oxidative damage caused by ethanol.

Two new steroids, 29-hydroperoxy-stigmasta-7,24(28)*E*-dien-3β-ol **(118)** and 24ξ-hydroperoxy-24-vinyl-lathosterol **(119)** (Figure 39), have been identified from the bark of *Melia azedarach* [103]. The structures of these steroids were elucidated using spectroscopic methods, including ^1^H NMR, ^13^C NMR, and HR ESI MS [103].

Among the cytotoxic evaluations, compounds **(118)** and **(119)** exhibited significant activity against the human lung cancer cell lines A549 and H460, as well as a human gastric carcinoma cell line, HGC27 [103]. The IC_50_ values ranged from 5.6 to 21.2 µg/mL, indicating a promising cytotoxic profile and highlighting their potential as anticancer agents [103]. Steroids such as 29-hydroperoxy-stigmasta-7,24(28)*E*-dien-3β-ol **(118)** and 24ξ-hydroperoxy-24-vinyl-lathosterol **(119)** are particularly noteworthy due to their novel structural elements, including hydroperoxy groups and vinyl substitutions at C-24.

Five new steroidal saponins were isolated from the fruits of *Tribulus terrestris,* and their structures were fully elucidated through spectroscopic and chemical analysis as (23S,25S)-5α-spirostane-24-one-3β,23-diol-3-O-{α-L-rhamnopyranosyl-(1→2)-O-[β-D-glucopyranosyl-(1→4)]-β-D-galactopyranoside} **(120)**, (24S,25S)-5α-spirostane-3β,24-diol-3-O-{α-L-rhamnopyranosyl-(1→2)-O-[β-D-glucopyranosyl-(1→4)]-β-D-galactopyranoside} **(121)**, 26-O-β-D-glucopyranosyl-(25R)-5α-furostan-2α,3β,22α,26-tetraol-3-O-{β-D-glucopyranosyl-(1→2)-O-β-D-glucopyranosyl-(1→4)-β-D-galactopyranoside} **(122)**, 26-O-β-D-glucopyranosyl-(25R)-5α-furostan-20(22)-en-2α,3β,26-triol-3-O-{β-D-glucopyranosyl-(1→2)-O-β-D-glucopyranosyl-(1→4)-β-D-galactopyranoside} **(123)**, and 26-O-β-D-glucopyranosyl-(25S)-5α-furostan-12-one-22-methoxy-3β,26-diol-3-O-{α-L-rhamnopyranosyl-(1→2)-O-[β-D-glucopyranosyl-(1→4)]-β-D-galactopyranoside} **(124)** (Figure 40) [104].

The cytostatic activity of compounds **(120–124)** was evaluated against HL-60 cancer cells, with IC_50_ values of 47.55 ± 6.7 μM, 46.87 ± 2.5 μM, 46.52 ± 4.7 μM, 49.45 ± 7.2 μM, and 41.42 ± 3.4 μM, respectively [104].

Five years later, another seven new steroidal saponins were isolated from *Tribulus terrestris*, including six furostanol saponins **(125–130)** and one spirostanol saponin **(131)** (Figure 41) [105]. Their structures were fully elucidated using 1D and 2D NMR spectroscopy, mass spectrometry, and chemical methods. These newly identified compounds were evaluated for their platelet aggregation activities. Compounds **(125)**, **(126,)** and **(128–130)** were tested for their effects on platelet aggregation induced by U46619, a TxA2 analog, in rat platelets. Tested compounds exhibited weak or no inhibitory effects on U46619-induced platelet aggregation, suggesting that these compounds do not significantly interfere with platelet activation [105].

### 3.2. Fungi

Two new lanostane triterpenoids **(132, 133)** and two new ergostane-type steroids **(134, 135)**, along with two known lanostane triterpenoids **(136, 137)** and one known steroid **(138)**, were isolated from the cultured mycelia of *Ganoderma capense* (CGMCC 5.71) (Figure 42) [106]. Their structure was determined by comprehensive spectroscopic analyses, including HR ESI-MS and 1D/2D NMR techniques. Ganoderma is a well-known medicinal macrofungus that is widely recognized for its diverse bioactivities, including antineoplastic and immunomodulatory properties [107,108,109,110,111,112,113,114]. Previous studies on *G*. *capense* have led to the discovery of novel sesquiterpenoids, motivating further exploration of its secondary metabolites, especially from cultured mycelia, which offer distinct physiological and environmental conditions compared with fruiting bodies [106].

Compound **(132)** displayed moderate cytotoxicity against the human cancer cell line NCI-H1650, with an IC_50_ value of 22.3 μM, while compound **(138)** exhibited cytotoxic activity against the HCT116 cell line, with an IC_50_ value of 17.4 μM [106]. Additionally, compounds **(133, 134)** and **(136, 137)** showed weak anti-HIV activity, with IC_50_ values of 23.5, 46.7, 21.6, and 30.1 μM, respectively. These results highlight the therapeutic potential of the newly identified compounds from *G. capense* [106].

The investigation of *Fomitiporia aethiopica*, a basidiomycete from East Africa, led to the discovery of five new pregnenolone-type steroids, named aethiopinolones A–E (compounds **(139–143)**, respectively) (Figure 43) [115]. These compounds were isolated from the fungal mycelial culture after fermentation on rice, followed by methanol extraction and chromatographic purification. The chemical structures of aethiopinolones A–E were determined with 1D and 2D NMR as well as HR MS data analysis [115].

Pregnenolone-type steroids are uncommon in fungal metabolism, but compounds **(139–140)** add to the growing list of steroids derived from the *Basidiomycota*, particularly within the *Hymenochaetales* order. While steroids are common in the *Basidiomycota*, they had not previously been reported from the genus *Fomitiporia*. Related compounds have been discovered in various fungus species, including *Phellinus igniarius* and the marine-derived *Phaeosphaeria spartinae*. Additionally, novel pregnenolone-like compounds have been identified in the genus *Fomitiporia* [115,116,117].

The biological activity of aethiopinolones A–E **(139–143)** was assessed for cytotoxicity against various mammalian cell lines, including MCF-7, A431, L929, and PC-3. Compounds **(141)** and **(142)** exhibited moderate cytotoxic effects across all tested cell lines [115]. Compound **(139)** showed the strongest cytotoxicity, with an IC_50_ of 8 μg/mL against PC-3 cells. Compound **(140)** displayed moderate activity only against the L929 and KB3.1 cell lines, with IC_50_ values of 45 μg/mL and 39 μg/mL, respectively. However, none of the compounds exhibited significant antimicrobial or nematocidal activities at tested concentrations (≤300 μg/mL and ≤100 μg/mL, respectively) [115].

A novel steroid (22*E*,24*R*)-6β-methoxyergosta-7,9(11),22-triene-3β,5α-diol **(144)** was identified from the Basidiomycota *Ganoderma sinense* as a potent natural inhibitor of hexokinase 2 (HK2) (Figure 44) [118]. HK2 is a key enzyme in the glycolysis pathway that is overexpressed in cancer cells and is a promising target for cancer therapy due to its critical role in tumor growth and metastasis [119,120,121,122,123].

The inhibition of HK2 has garnered significant attention in cancer research due to its critical role in tumor metabolism. HK2 inhibition disrupts the glycolytic pathway, which is crucial for cancer cell survival and proliferation. Studies have shown that HK2 inhibition reduces glucose uptake, decreases lactate production, and impairs tumor growth [124]. In pancreatic cancer, where HK2 is notably overexpressed, inhibition of this enzyme has been associated with decreased anchorage-independent growth and reduced invasive capabilities [125].

Using structure-based virtual ligand screening, compound **(144)** demonstrated high binding affinity to HK2. This finding was confirmed through microscale thermophoresis (MST), enzyme inhibition assays, and cell-based experiments. Compound **(144)** inhibited HK2 with an IC_50_ of 2.06 μM, acting as a non-competitive inhibitor [118]. It reduced the maximal reaction rate (V_max_) of glucose without significantly affecting the binding affinity (Km). This mode of inhibition is particularly valuable as it allows for the modulation of enzyme activity without directly competing with the substrate, potentially leading to more selective targeting of cancer cells. The inhibition of HK2 has been shown to effectively disrupt cancer cell metabolism, offering a potential avenue for therapeutic intervention across various cancer types [118].

## 4. Conclusions

Over the past two decades, research on marine and terrestrial steroids has expanded significantly, revealing remarkable structural diversity and bioactivity. This review highlights that approximately 60–70% of the newly discovered steroids originate from marine sources, reinforcing the role of marine ecosystems as reservoirs of bioactive compounds.

Novel steroidal frameworks, particularly those featuring unique hydroxylation, sulfation, and glycosylation patterns, have demonstrated potent anticancer, antimicrobial, antidiabetic, and immunomodulatory properties. Marine-derived polyhydroxylated steroids have shown cytotoxicity comparable to clinically used drugs, with some compounds exhibiting additional antibacterial and enzyme-inhibitory activities. Notable examples include steroids from *Acanthaster planci* and *Dendrodoris fumata*, which have shown promising effects against cancer cells and pathogenic bacteria. Terrestrial sources have also contributed significant steroidal compounds, such as chenisterol from *Chenopodium badachschanicum*, which exhibited broad-spectrum antimicrobial activity.

Despite these advances, several challenges remain in translating bioactive steroids into clinical applications. Many marine-derived steroids suffer from poor bioavailability and rapid metabolism, limiting their therapeutic potential. Additionally, sustainable access to rare natural steroids poses a challenge, necessitating the development of synthetic and semi-synthetic approaches. Advances in cheminformatics, enzymatic modifications, and AI-driven molecular design offer promising solutions to optimize steroidal frameworks for enhanced efficacy and stability.

Future research should prioritize structure–activity relationship studies and targeted modifications to refine steroidal scaffolds. Synthetic biology and engineered biosynthesis hold great potential for generating complex steroidal architectures with improved bioactivity. The integration of computational modeling, high-throughput screening, and rational drug design will be essential for accelerating the discovery of steroid-based therapeutics.

By merging natural product research with modern synthetic and computational methods, the full pharmacological potential of steroids can be unlocked. This integrated approach will be crucial to addressing global health challenges such as antimicrobial resistance and cancer, paving the way for next-generation steroid-based pharmaceuticals. The diversity and potency of the newly discovered steroids underscore the need for continued exploration and investment in this field.

## Figures and Tables

**Figure 1 ijms-26-03203-f001:**
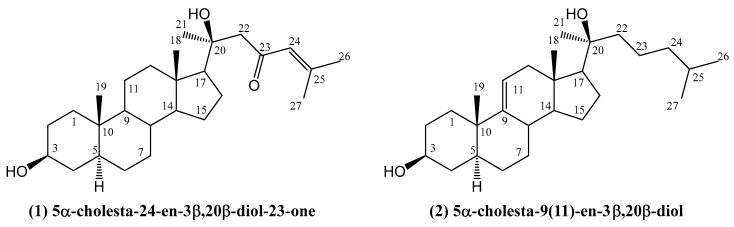
Two new steroids isolated from *Acanthaster planci*.

**Figure 2 ijms-26-03203-f002:**
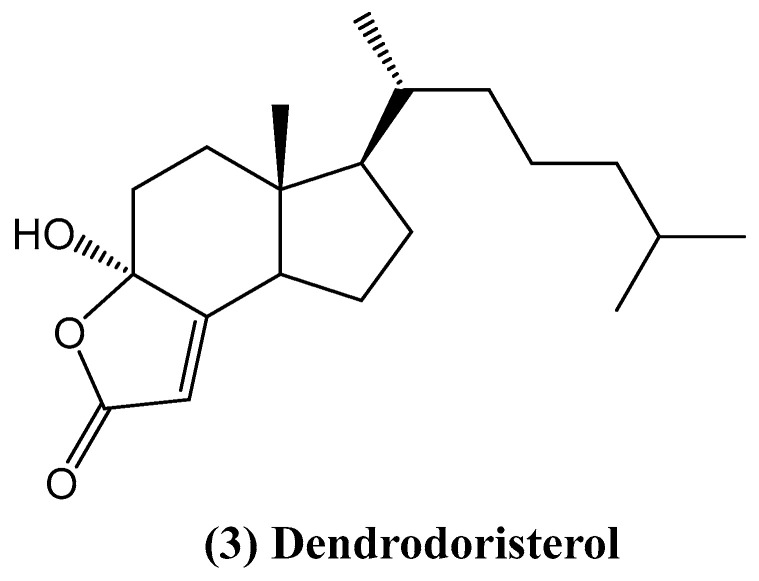
Structure of dendrodoristerol.

**Figure 3 ijms-26-03203-f003:**
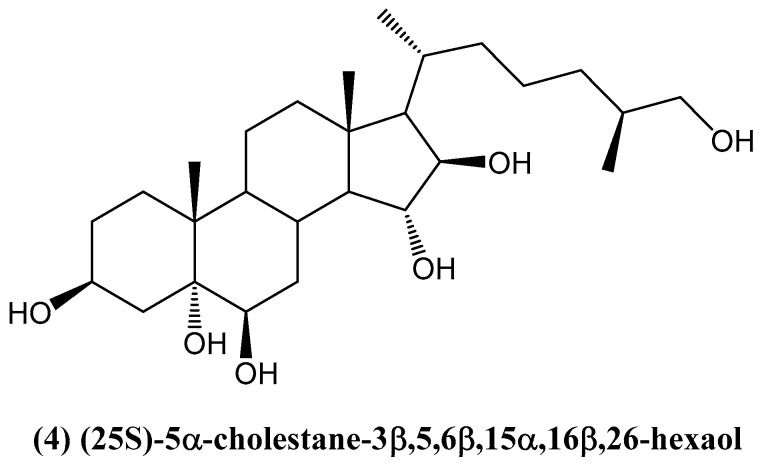
Structure of new steroid extracted from *Ctenodiscus crispatus*.

**Figure 4 ijms-26-03203-f004:**
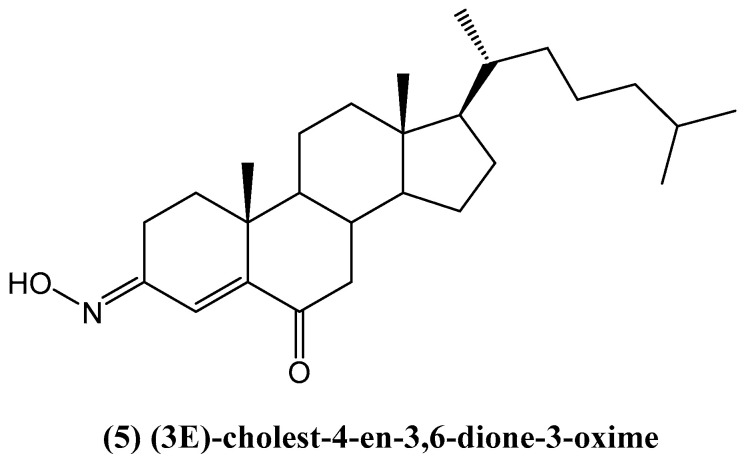
Structure of new steroid oxime isolated from *Cinachyrella australiensis*.

**Figure 5 ijms-26-03203-f005:**
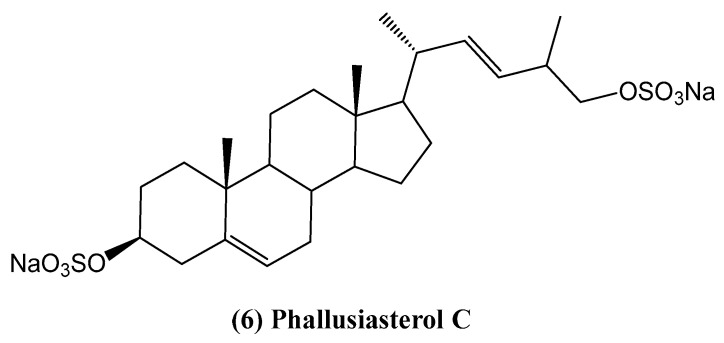
Structure of the steroid isolated from *Phallusia fumigate*.

**Figure 6 ijms-26-03203-f006:**
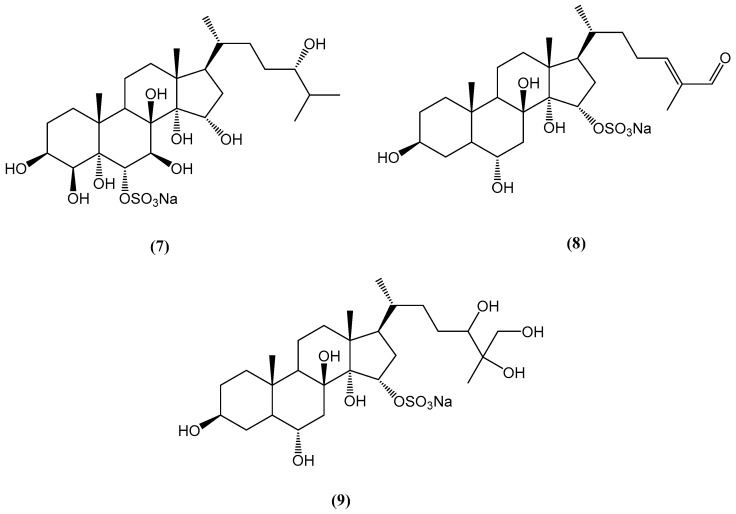
Chemical structures of highly hydroxylated steroids from *Archaster typicus*.

**Figure 7 ijms-26-03203-f007:**
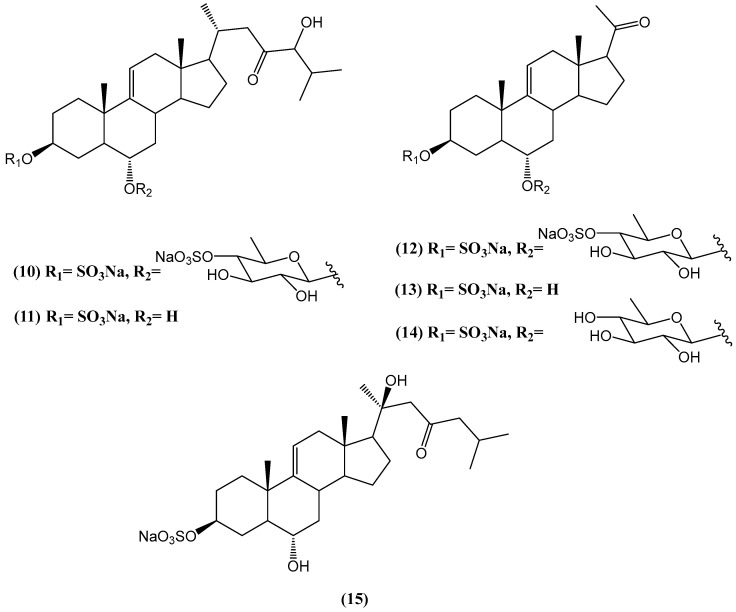
Chemical structures of steroids from *Aphelasterias japonica*.

**Figure 8 ijms-26-03203-f008:**
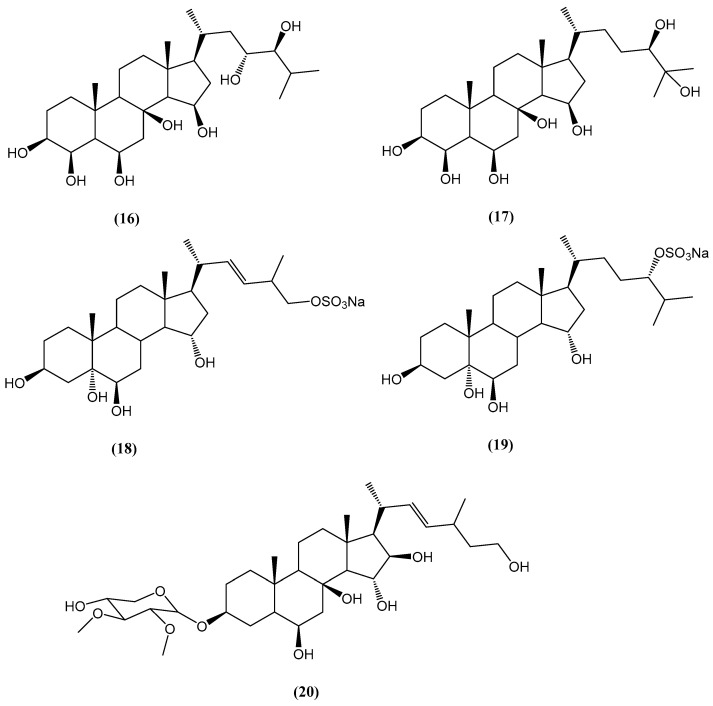
Chemical structures of polyhydroxysteroids from *Henricia leviuscula*.

**Figure 9 ijms-26-03203-f009:**
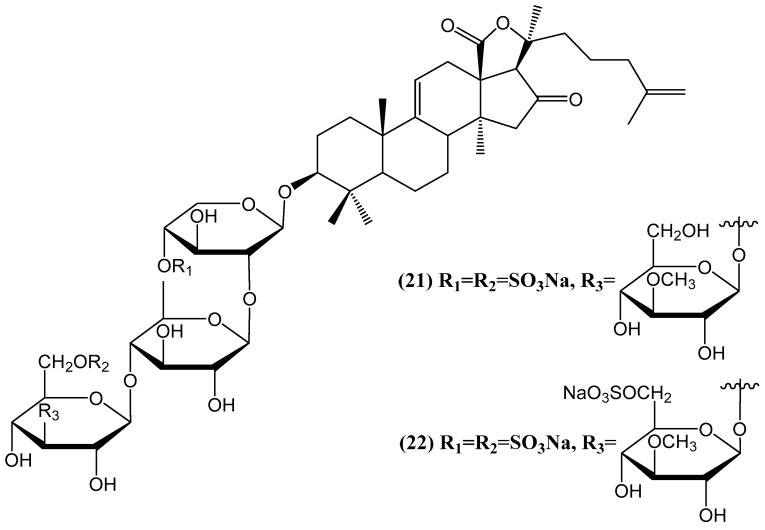
Chemical structures of sulfated triterpene glycosides from the Patagonian sea cucumber *Hemoiedema spectabilis*.

**Figure 10 ijms-26-03203-f010:**
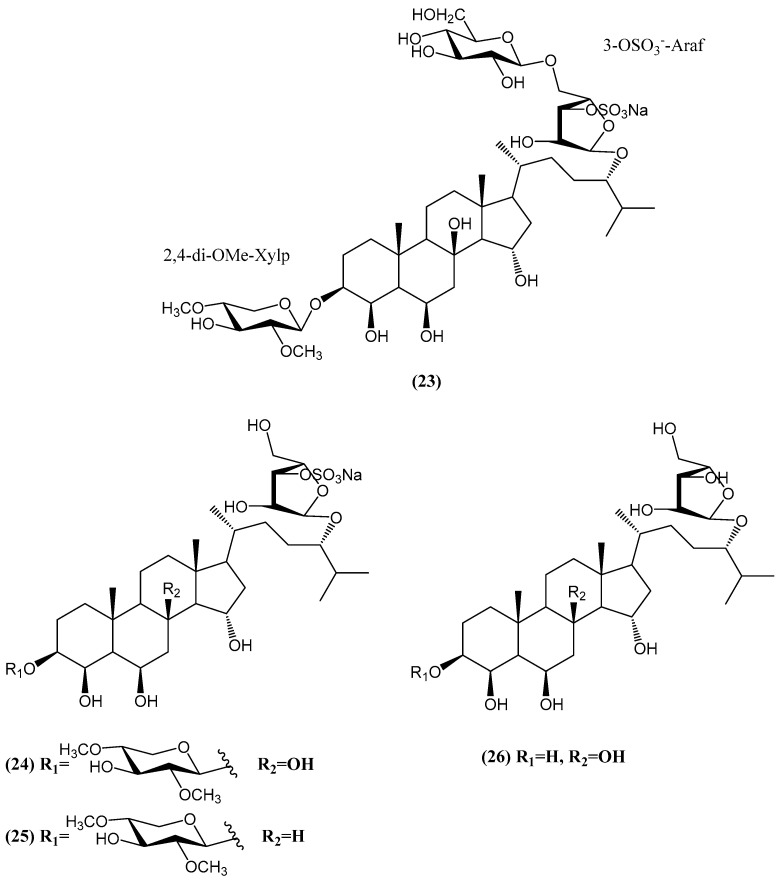
Spiculiferosides A **(23)**, B **(24)**, C **(25)**, and D **(26)** isolated from the starfish *Henricia leviuscula spiculifera*.

**Figure 11 ijms-26-03203-f011:**
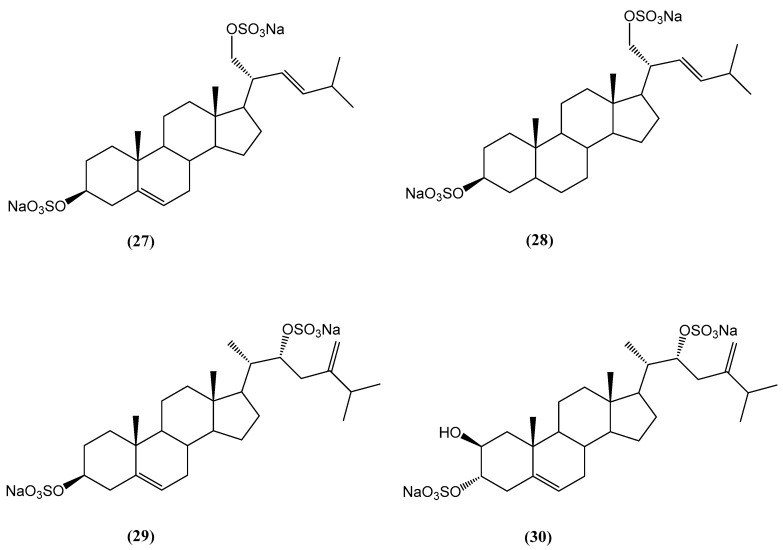
Chemical structures of steroids isolated from *Pteraster marsippus*.

**Figure 12 ijms-26-03203-f012:**
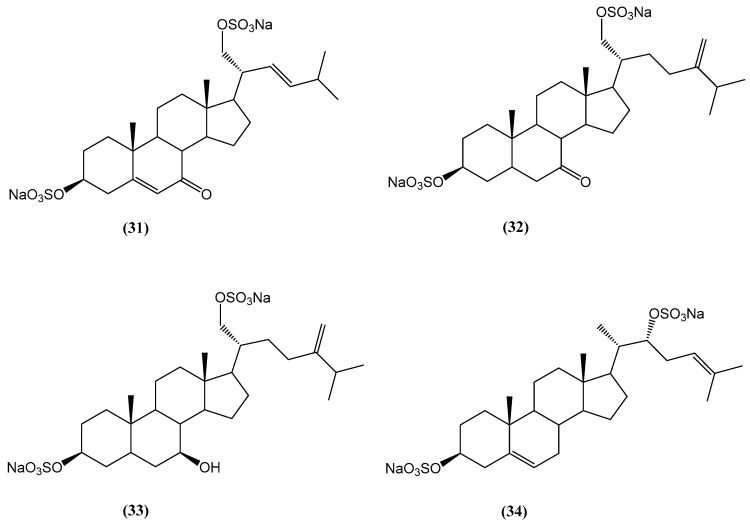
Chemical structures of new disulfated steroids from *Pteraster marsippus*.

**Figure 13 ijms-26-03203-f013:**
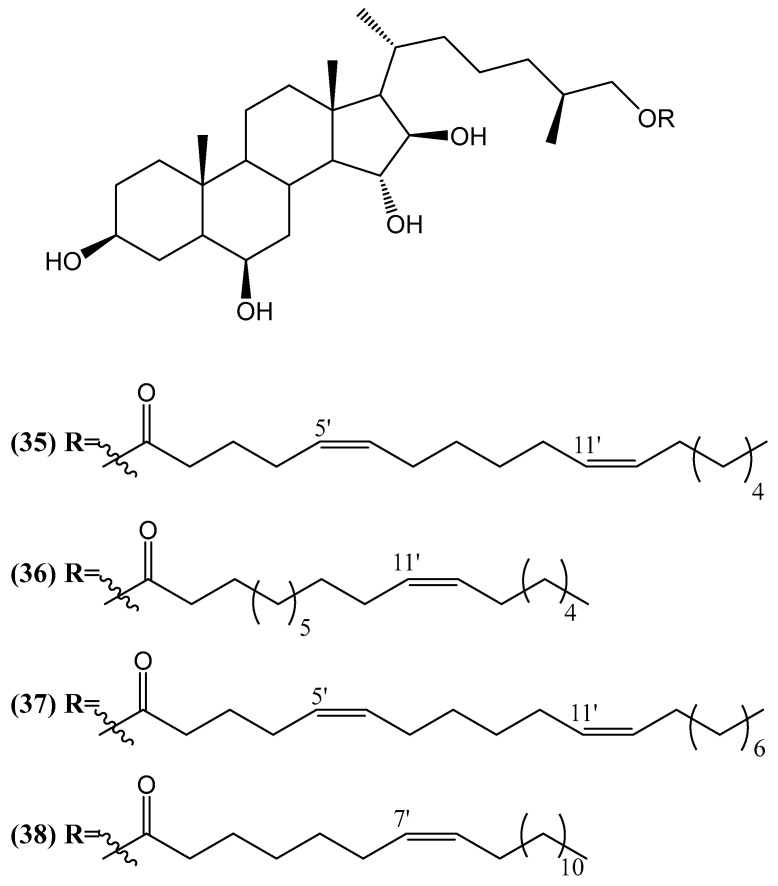
New conjugates of polyhydroxysteroids with long-chain fatty acids from the Starfish *Ceramaster patagonicus*.

**Figure 14 ijms-26-03203-f014:**
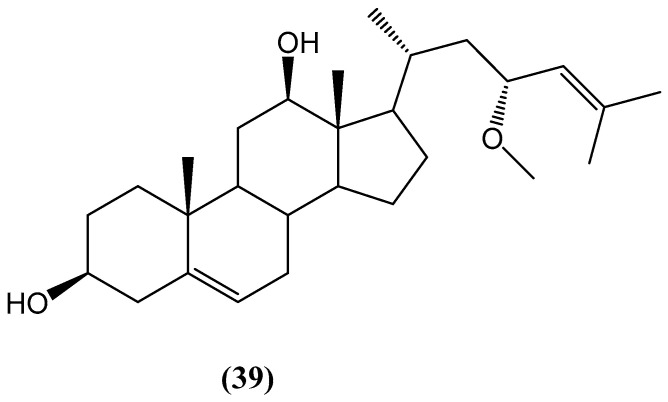
(23R)-methoxycholest-5,24-dien-3β-ol isolated from *Cryptosula pallasiana*.

**Figure 15 ijms-26-03203-f015:**
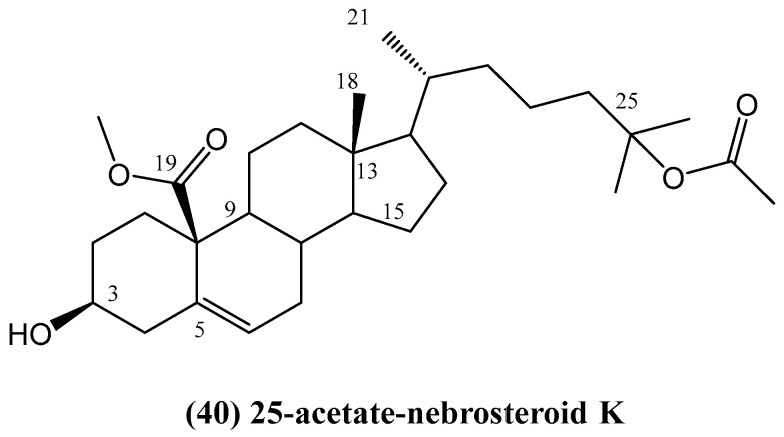
The new 19-oxygenated steroid from *Dichotella gemmacea*.

**Figure 16 ijms-26-03203-f016:**
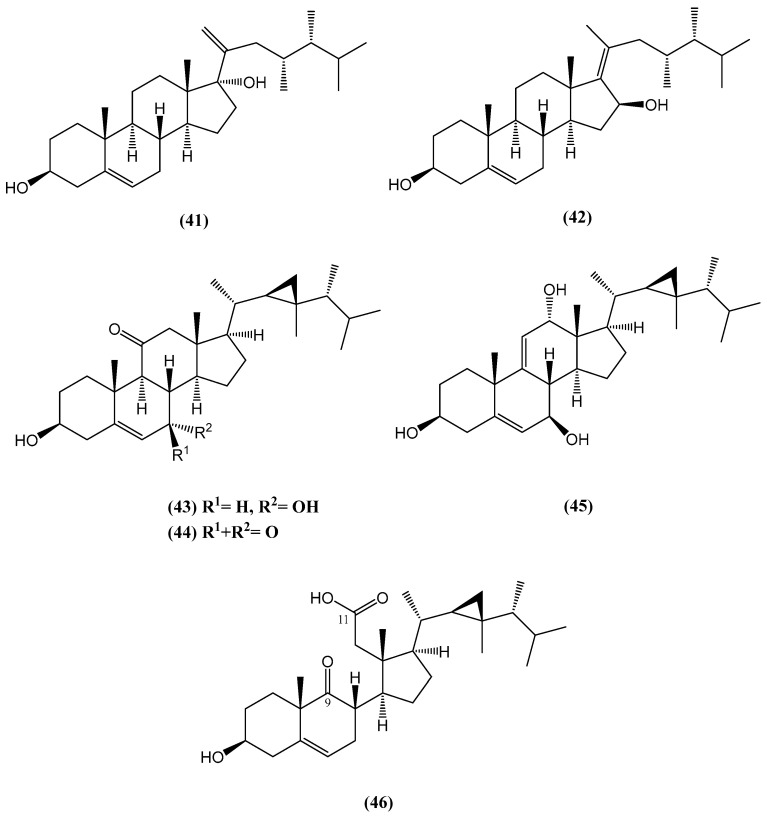
New steroids extracted from the coral *Klyxum flaccidum*.

**Figure 17 ijms-26-03203-f017:**
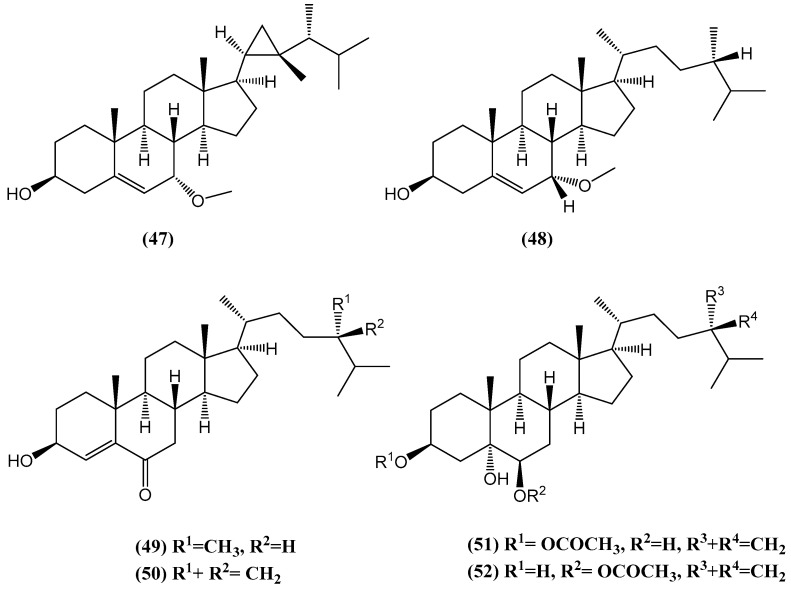
Structure of new steroids isolated from the methanol extract of *Sinularia*.

**Figure 18 ijms-26-03203-f018:**
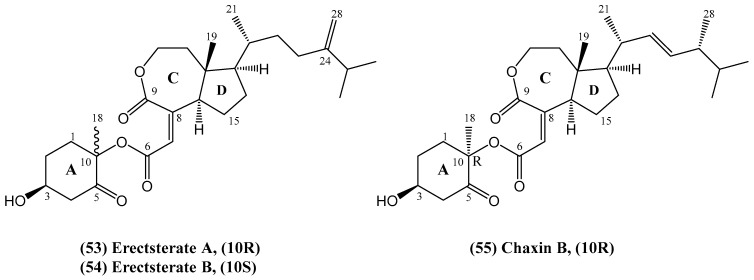
Structures of the erectsterates (A and B) and Chaxin B from *Sinularia erecta*.

**Figure 19 ijms-26-03203-f019:**
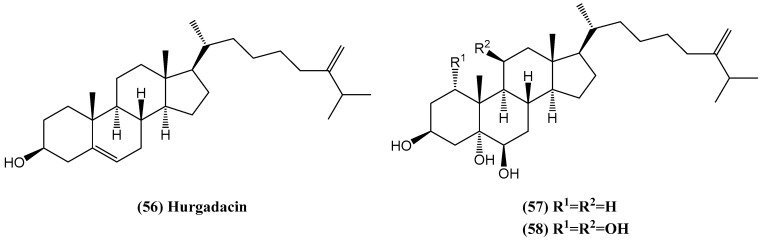
Steroids found in the soft coral *Sinularia polydactyla*.

**Figure 20 ijms-26-03203-f020:**
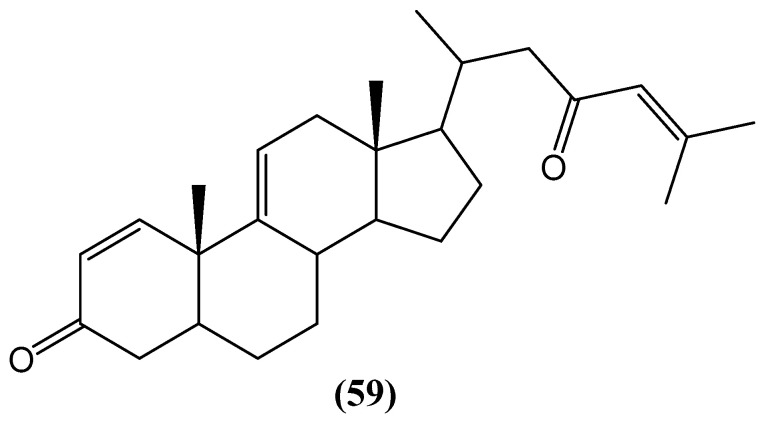
Cytotoxic ketosteroids from the Red Sea soft coral *Dendronephthya* sp.

**Figure 21 ijms-26-03203-f021:**
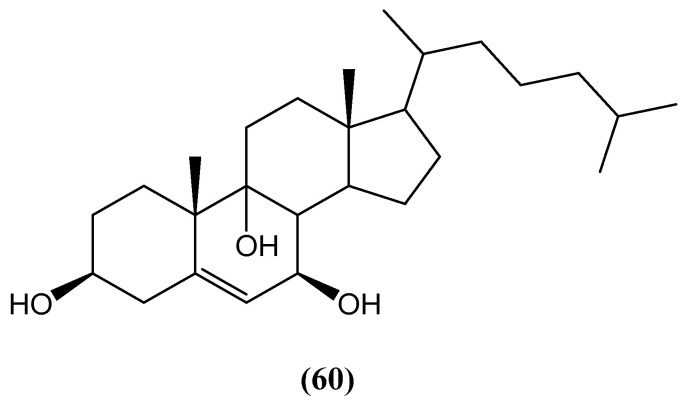
New steroid from *Petrosia* sp.

**Figure 22 ijms-26-03203-f022:**
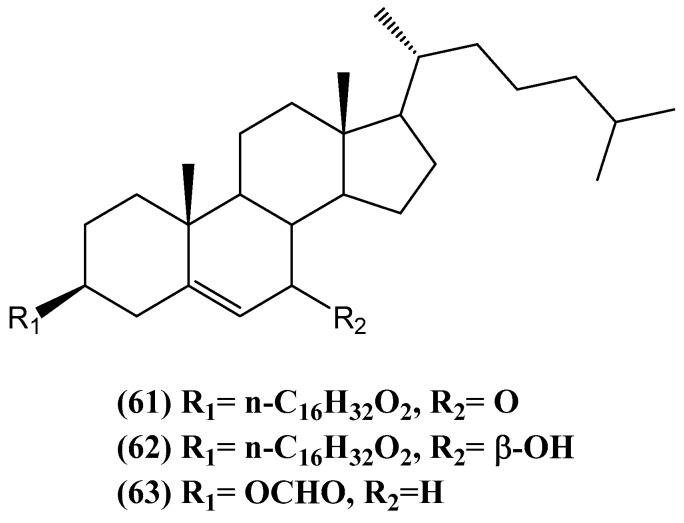
Steroids isolated from *Antipathes dichotoma*.

**Figure 23 ijms-26-03203-f023:**
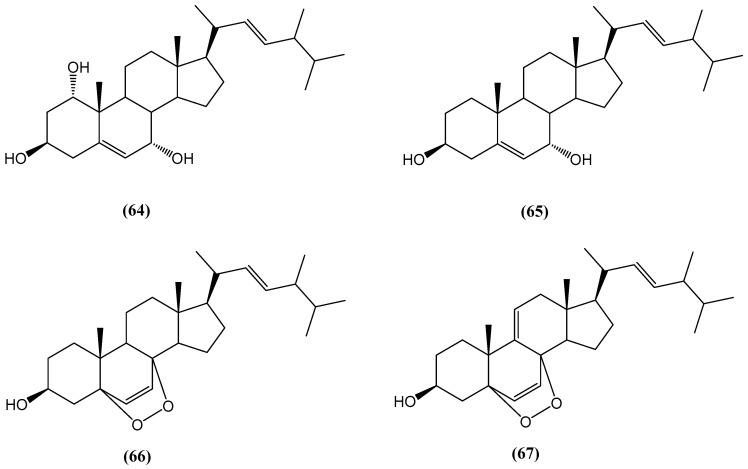
Steroids extracted from the black coral *Antipathes dichotoma*.

**Figure 24 ijms-26-03203-f024:**
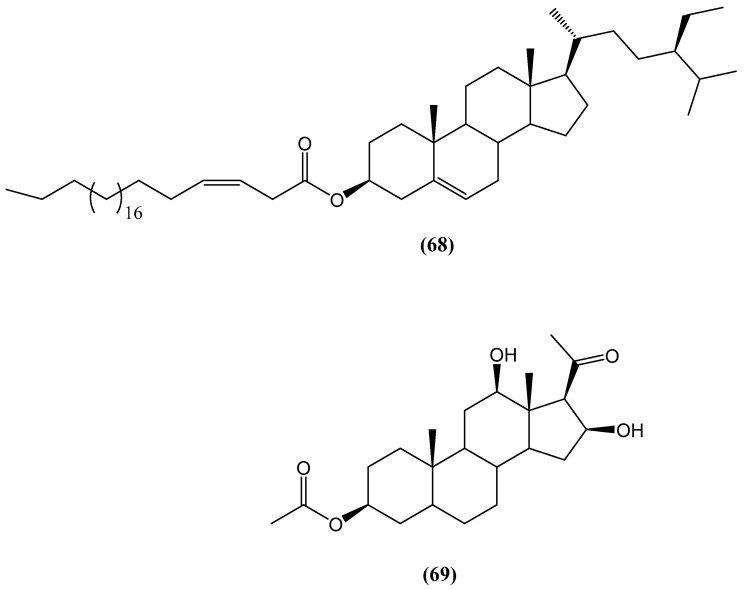
Steroid compounds from *Echinoclathria gibbosa*.

**Figure 25 ijms-26-03203-f025:**
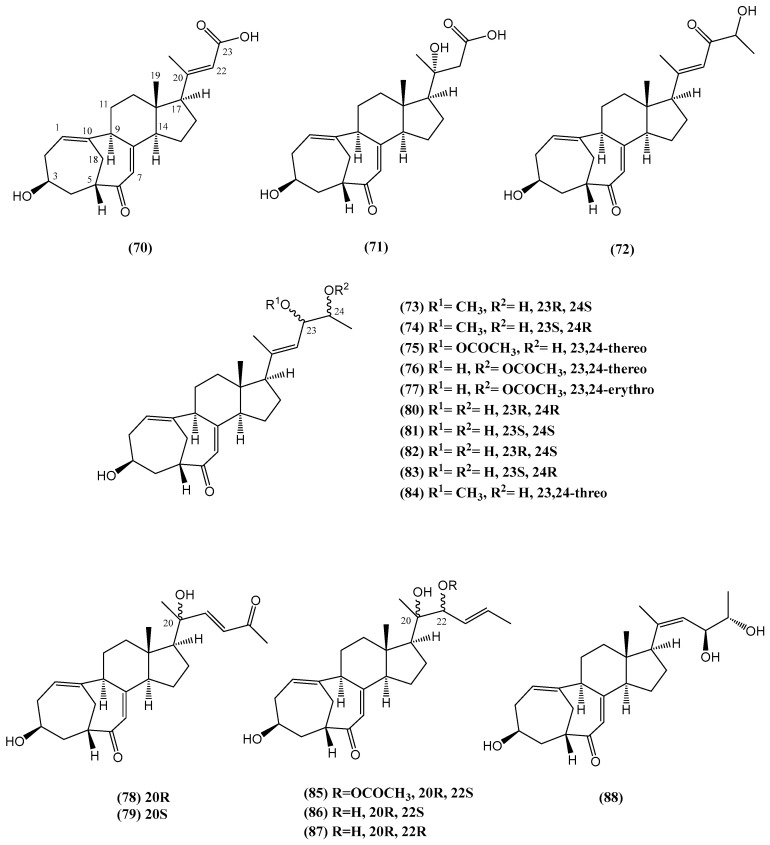
Steroids extracted from the deep-sea-derived fungus *Rhizopus* sp. W23.

**Figure 26 ijms-26-03203-f026:**
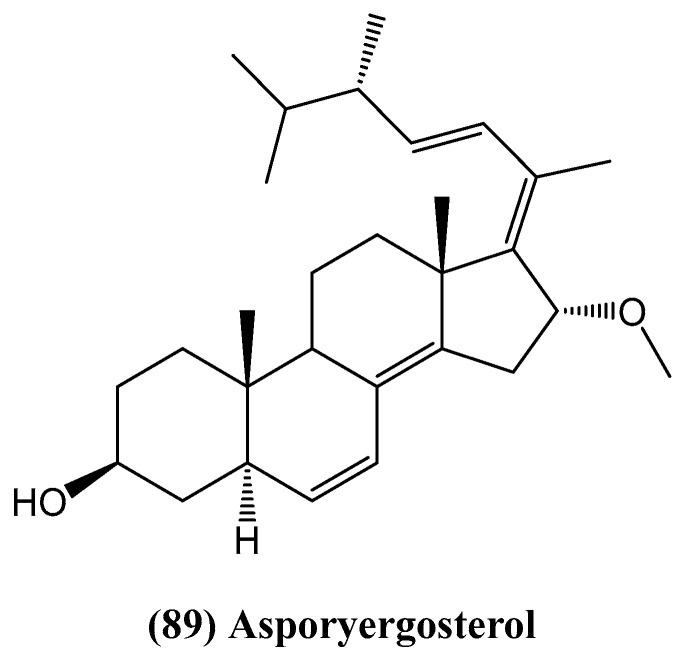
Structure of asporyergosterol.

**Figure 27 ijms-26-03203-f027:**
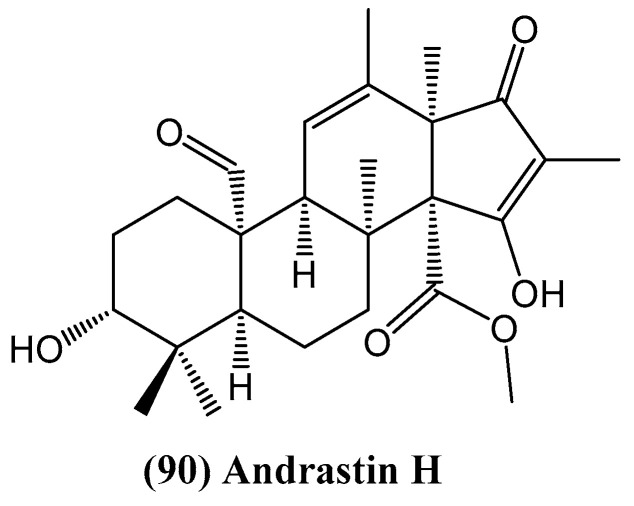
Structure of andrastin H.

**Figure 28 ijms-26-03203-f028:**
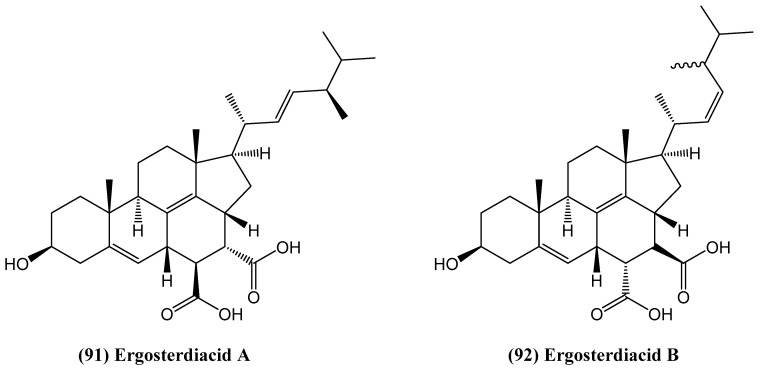
Structure of ergosterdiacids A and B.

**Figure 29 ijms-26-03203-f029:**
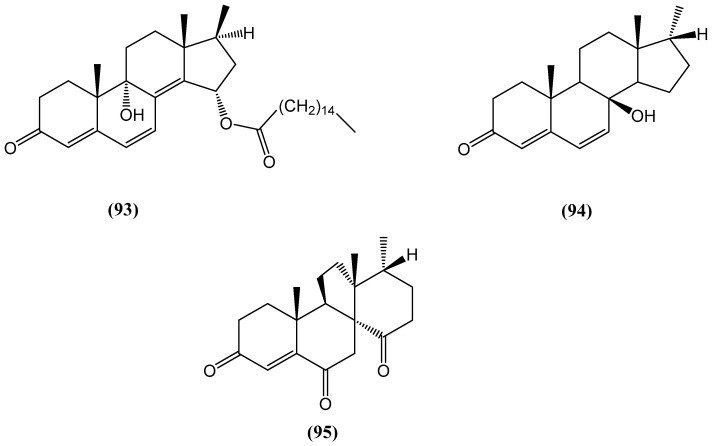
Structure of steroids extracted from *Penicillium oxalicum* HL-44, isolated from *Sinularia gaweli*.

**Figure 30 ijms-26-03203-f030:**
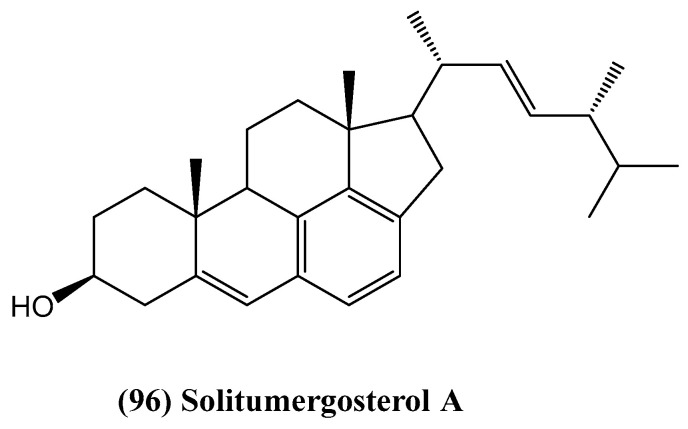
Structure of Solitumergosterol A.

**Figure 31 ijms-26-03203-f031:**
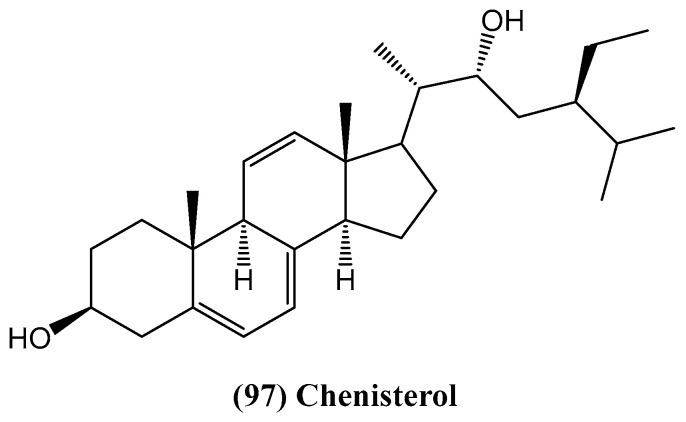
Structure of chenisterol.

**Figure 32 ijms-26-03203-f032:**
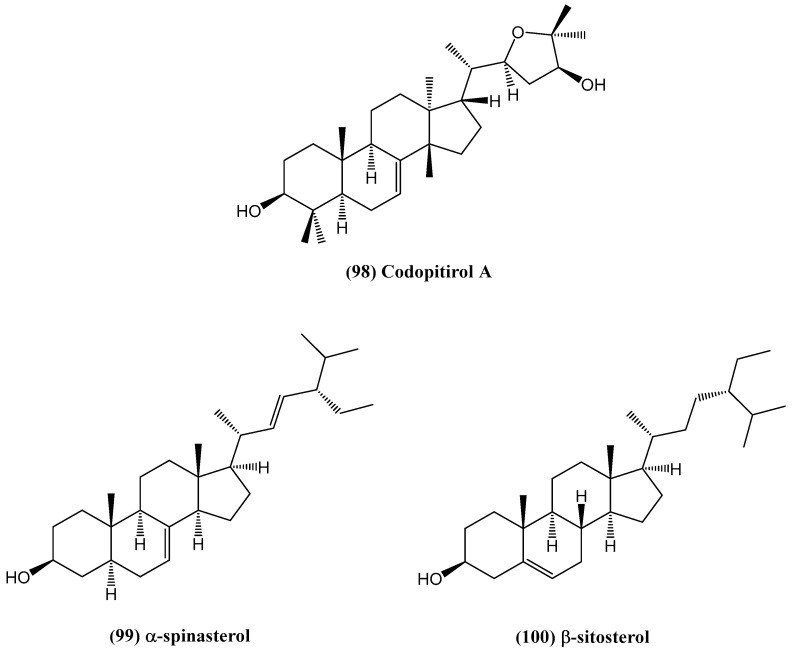
Structures of selected steroids from *Codonopsis pilosula* var. *modesta*.

**Figure 33 ijms-26-03203-f033:**
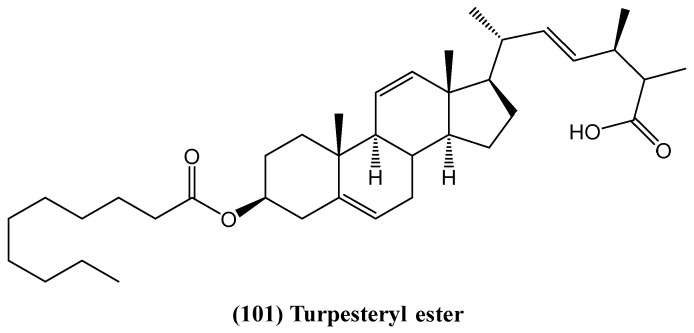
Structure of turpesteryl ester from *Ipomoea turpethum* (L.) R.Br.

**Figure 34 ijms-26-03203-f034:**
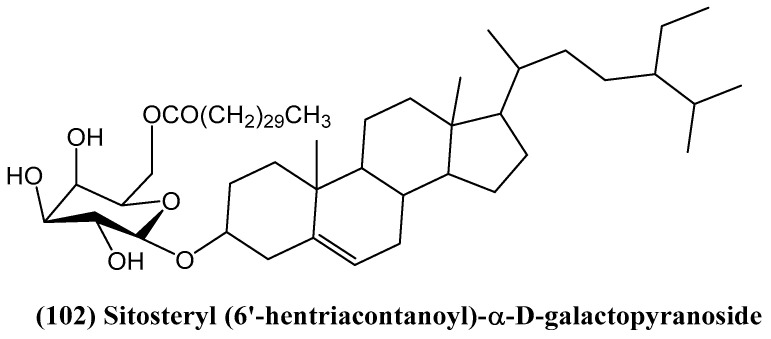
New steroid glycoside from *Cyperus rotundus* L.

**Figure 35 ijms-26-03203-f035:**
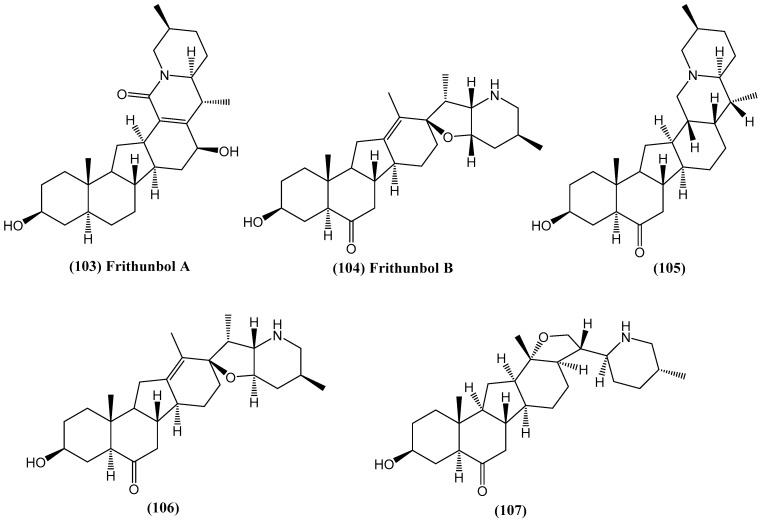
Structures of compounds isolated from *Fritillaria thunbergia*.

**Figure 36 ijms-26-03203-f036:**
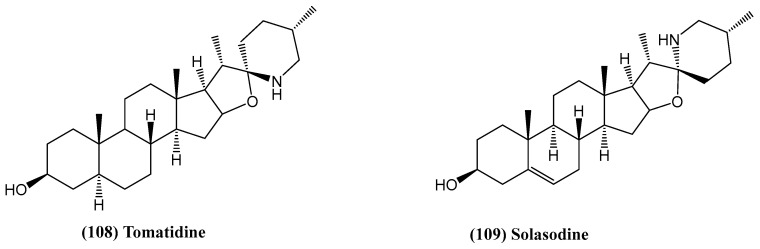
Chemical structures of tomatidine and solasodine.

**Figure 37 ijms-26-03203-f037:**
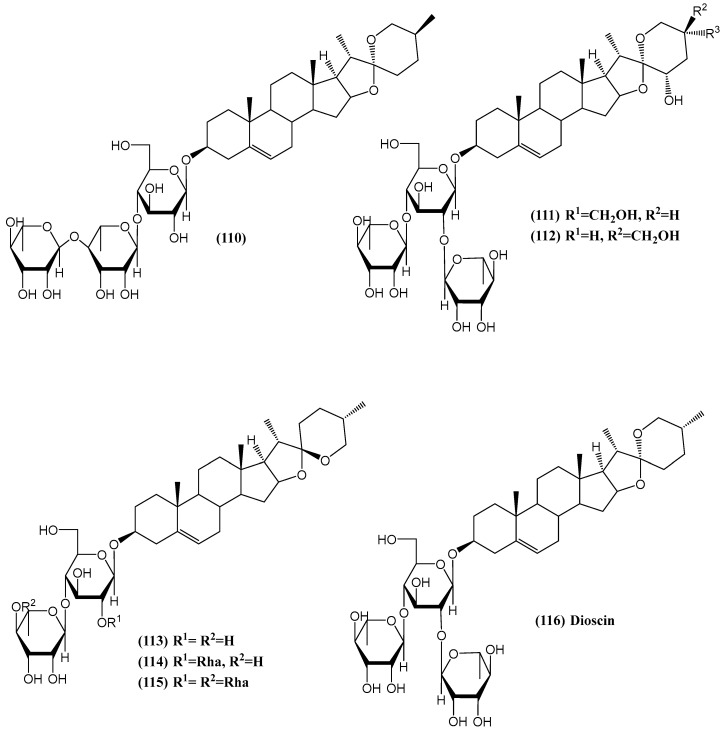
Active compounds from *Borassus flabellifer*.

**Figure 38 ijms-26-03203-f038:**
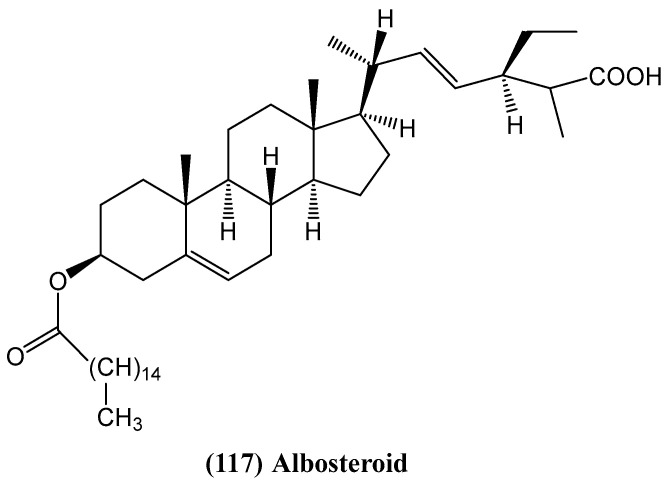
Structure of new steroid isolated from *Morus alba*.

**Figure 39 ijms-26-03203-f039:**
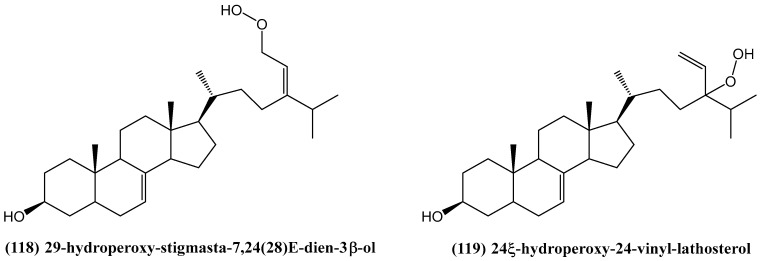
Structures of new steroids from *Melia azedarach*.

**Figure 40 ijms-26-03203-f040:**
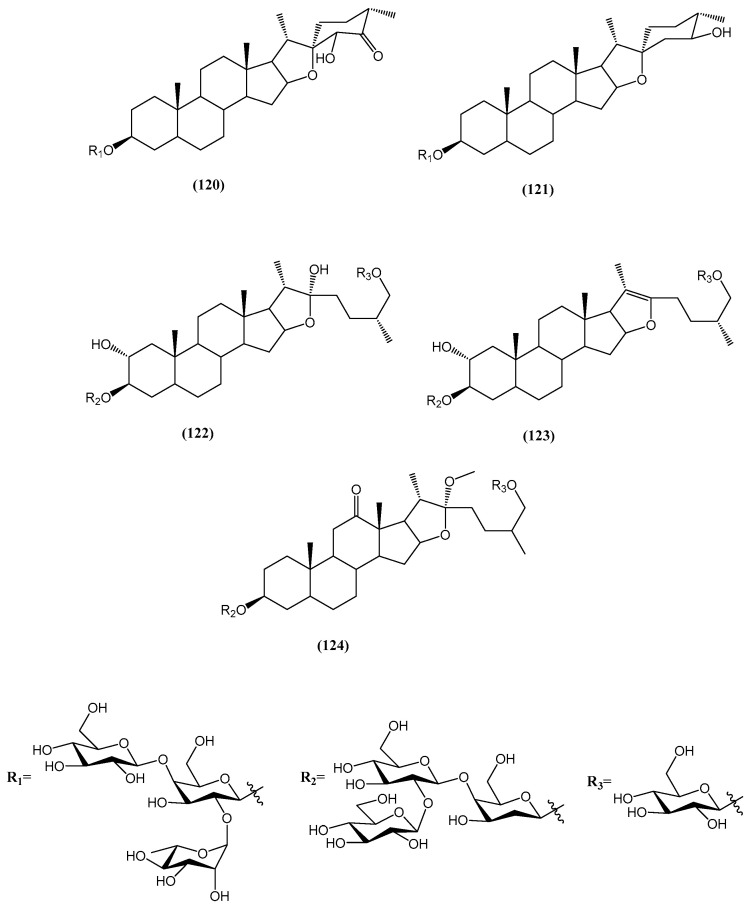
Structures of new steroids from the fruits of *Tribulus terrestris*.

**Figure 41 ijms-26-03203-f041:**
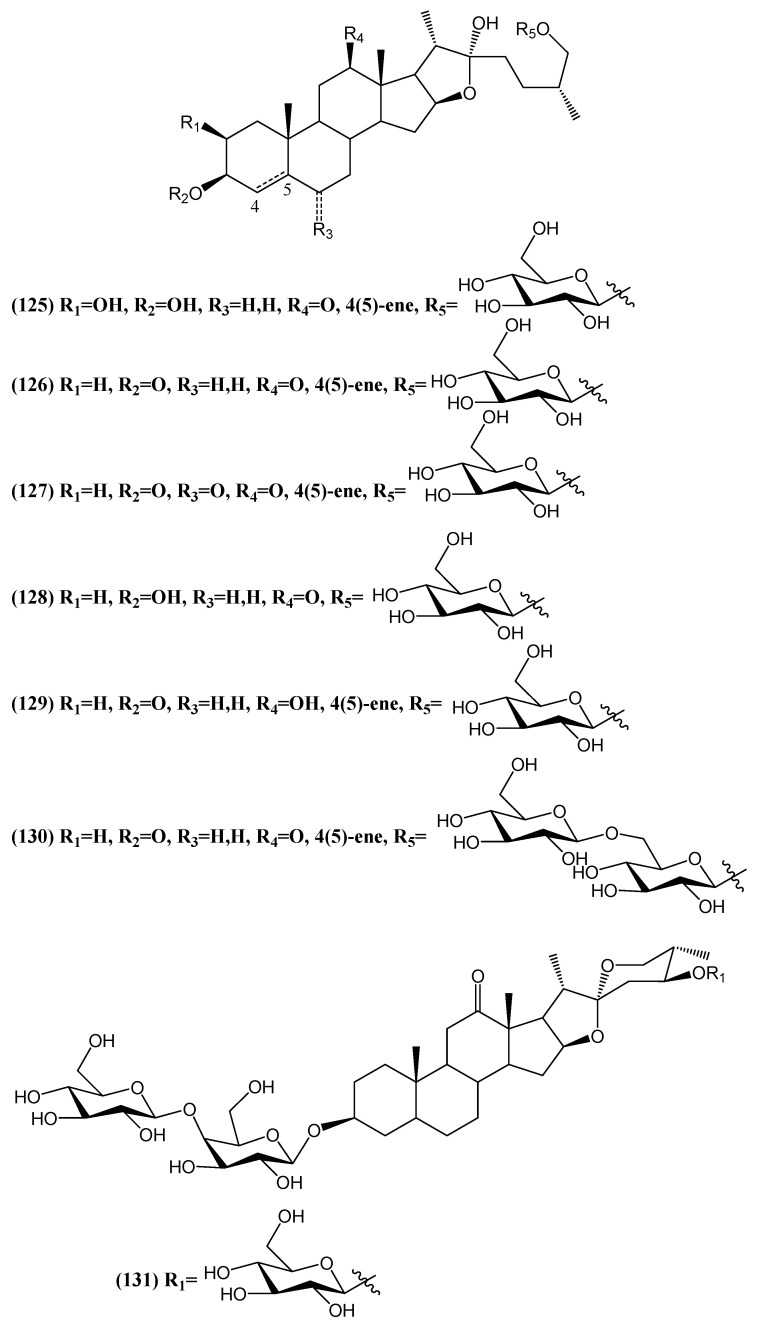
Structures of new steroidal saponins from the fruits of *Tribulus terrestris*.

**Figure 42 ijms-26-03203-f042:**
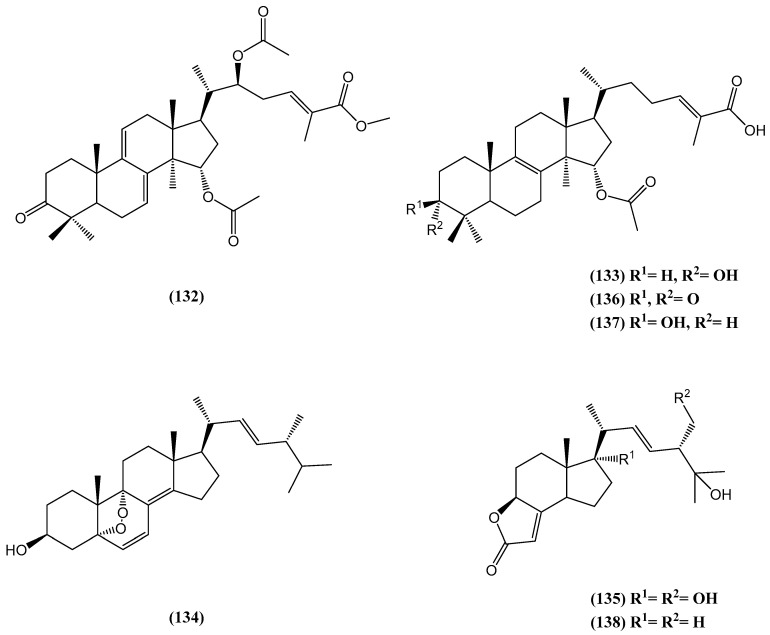
Steroids isolated from cultured mycelia of *Ganoderma capense* (CGMCC 5.71).

**Figure 43 ijms-26-03203-f043:**
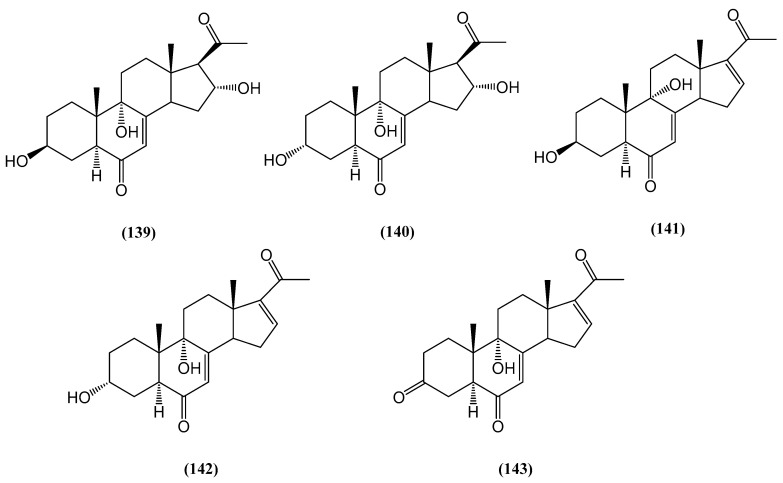
New aethiopinolones from *Fomitiporia aethiopica*.

**Figure 44 ijms-26-03203-f044:**
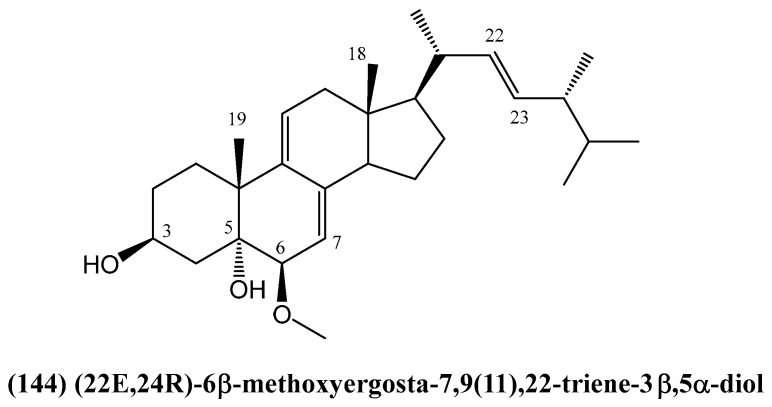
Structure of the steroid from *Ganoderma sinense*.

## Data Availability

This review is based on the data available in the cited publications. No new data were generated or analyzed in this study.

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
