# Peer review of "Recent Advances in Steroid Discovery: Structural Diversity and Bioactivity of Marine and Terrestrial Steroids"

_ijms, 2025, doi:10.3390/ijms26073203_

Round 1
Reviewer 1 Report
Comments and Suggestions for Authors
The authors review the discoveries of new steroids over the past two decades, highlighting the structural diversity of steroids from marine and terrestrial sources, including marine invertebrates, fungi, and medicinal plants. These compounds exhibit a range of biological activities, such as anticancer, antimicrobial, antidiabetic, and anti-inflammatory properties.
Key points include the identification of novel steroids from various natural sources, such as marine sponges, fungi, and medicinal plants, and the potential of steroids in addressing global health challenges like antibiotic resistance and cancer.
However, one of the main problems with the review comes from the main title: “Last two decades of steroid discovery”. The reader expects to find a review describing the research about steroids during the last twenty years integrating that into the previous knowledge.
The authors only indicate some completely new steroid molecules from some sources without a complete view of these in the steroidal lipids group. Moreover, they focused only on some examples without showing the whole perspective of the proposal in the title, for the last twenty years.
Taking this into account, this referee misses information on other articles that are also cited, such as:
Pang, C.; Chen, Y.-H.; Bian, H.-H.; Zhang, J.-P.; Su, L.; Han, H.; Zhang, W. Anti-Inflammatory Ergosteroid Derivatives from the Coral-Associated Fungi Penicillium oxalicum HL-44. Molecules 2023, 28, 7784. https://doi.org/10.3390/molecules28237784
He, Z.-H.; Xie, C.-L.; Hao, Y.-J.; Xu, L.; Wang, C.-F.; Hu, M.-Y.; Li, S.-J.; Zhong, T.-H.; Yang, X.-W. Solitumergosterol A, a unique 6/6/6/6/5 steroid from the deep-sea-derived Penicillium solitum MCCC 3A00215. Org. Biomol. Chem. 2021, 19, 9369–9372.
Wang, Y.; Chen, W.; Xu, Z.; Bai, Q.; Zhou, X.; Zheng, C.; Bai, M.; Chen, G. Biological secondary metabolites from the Lumnitzera littorea-derived fungus Penicillium oxalicum HLLG-13. Mar. Drugs 2023, 21, 22.
Krohn, K.; Biele, C.; Aust, H.J.; Draeger, S.; Schulz, B. Herbarulide, a ketodivinyllactone steroid with an unprecedented homo-6-oxaergostane skeleton from the endophytic fungus Pleospora herbarum. J. Nat. Prod. 1999, 62, 629–630.
Qiu, Y.; Chen, S.; Yu, M.; Shi, J.; Liu, J.; Li, X.; Chen, J.; Sun, X.; Huang, G.; Zheng, C. Natural Products from Marine-Derived Fungi with Anti-Inflammatory Activity. Mar. Drugs 2024, 22, 433. https://doi.org/10.3390/md22100433
Liu, Z.; Dong, Z.; Qiu, P.; Wang, Q.; Yan, J.; Lu, Y.; Wasu, P.A.; Hong, K.; She, Z. Two new bioactive steroids from a mangrove-derived fungus Aspergillus sp. Steroids 2018, 140, 32–38.
Huong PTM, Phong NV, Thao NP, Binh PT, Thao DT, Thanh NV, Cuong NX, Nam NH, Thung DC, Minh CV. Dendrodoristerol, a cytotoxic C20 steroid from the Vietnamese nudibranch mollusk Dendrodoris fumata. J Asian Nat Prod Res. 2020 Feb;22(2):193-200. doi: 10.1080/10286020.2018.1543280
and others.
Moreover, considering this is a review, references 24-27 have been published between 2016-19, thus, inside the time considered by the authors (last two decades). Are they not relevant to the topic? Reference 65, a publication of 2009, where are these methylated steroids?
Another main problem with this review is the lack of organization in the data description. I propose to organize the information into different sections dividing section 2 according to some specific characteristics of the shown steroids (source, grouping those from starfishes, corals, fungi, and plants in separate sections, or classifying according to the physiological effects, cytotoxicity, hypoglycemics, bactericides…). This will contribute to a more agile reading and comparison of the molecules according to some characteristics. Moreover, when indicating the isolation of a new steroid with other potentially related compounds, describe all the compounds (including these in the figures if necessary) to give a wider view of the compounds and the chemical environment of these compounds.
Another point comes from the use of IC50, LD50, and ED50. From my point of view, IC50 describes the concentration needed to inhibit a biological activity (i.e., an enzyme), while LC50 refers to the concentration that causes lethality to half the population. Thus, sometimes IC50 is used when LD50 seems more appropriate. In other cases, the whole meaning of the biological effect of the compounds looks confusing.
Some sentences are unnecessary and redundant (i.e. lines 338-339, “This is the first report…” this is indicated at the beginning of the previous paragraph; lines 404-405, this information is included in the previous paragraph). Sometimes (i.e., lines 222-225), the sentence could be rewritten in a more fluid style: “Two new steroids X and Y (37,38; Fig. 10) have been identified from the bark of Melia azedarach [55].” For me, lines 304-308 are superfluous.
Some references should be indicated, i.e. line 274 “Previous studies…” I miss the reference.
I do not understand lines 294-295: “…, discovering pregnenolone-like compounds in Formitiporia novel [62-63]”.
Line 53, Echinoderms should not be in italics.
Some lines (69, 127 and others) are not aligned with the text.
Line 87: should it be read as “Dendrodoristerol is a degraded sterol”?
Lines 114-115: “characterized by minor modifications”, please, indicate which these modifications are.
Line 163, “this study reports…” What study? Reference?
Line 165, algicolous should not be in italics.
Lines 438-440. Contextualize with the other starfishes.
Lines 496-498. The reader does not have the context about that. This is the first (and only) reference to a biosynthetic pathway in all the compounds cited in the text.
Some references should be reviewed, i.e. reference 54 and reference 67.
Author Response
Comment: The title “Last two decades of steroid discovery” suggests a comprehensive integration of recent steroid research into the broader field of steroidal lipids. However, the review only highlights certain newly discovered steroid molecules without providing a complete perspective on their relationship to the broader steroidal lipid group over the past twenty years.
Response: The title has been revised to more accurately reflect the scope of the review. Additionally, the manuscript has been reorganized to categorize steroids based on their natural sources, ensuring a more structured presentation. The conclusion section has also been expanded to better contextualize recent findings within the broader landscape of steroidal research.
Comment: The review omits discussion of some cited articles, including Pang et al. (2023), He et al. (2021), Wang et al. (2023), and others.
Response: The manuscript has been updated to incorporate the suggested references where relevant. These studies have been integrated into the appropriate sections, ensuring a more comprehensive discussion of recent steroid discoveries.
Comment: Some references (24–27) fall within the covered timeframe (last two decades). Their relevance should be clarified. Additionally, the omission of methylated steroids from reference 65 raises concerns.
Response: The references in question have been reviewed, and relevant studies have been retained and incorporated where appropriate. Reference 65 has been removed, as its relevance to the current review was determined to be limited.
Comment: The review lacks organization in data description. Section 2 should be divided into subsections based on either the source of steroids (e.g., starfishes, corals, fungi, plants) or their biological activity (e.g., cytotoxic, hypoglycemic, bactericidal).
Response: The manuscript has been reorganized to classify steroids based on their natural sources, providing a clearer structure and improving readability. This revision facilitates easier comparison of steroid compounds according to their origins and associated bioactivities.
Comment: The discussion of newly isolated steroids should include all related compounds from the same study, along with relevant figures where necessary, to provide a broader chemical context.
Response: Additional related compounds have been described, and figures have been updated where appropriate to offer a more comprehensive overview of newly isolated steroids and their structural context.
Comment: The use of IC50, LD50, and ED50 appears inconsistent. IC50 refers to inhibition of biological activity, whereas LD50 pertains to lethality. In some cases, IC50 is used when LD50 seems more appropriate.
Response: The terminology used in the manuscript follows the original terminology as presented in the referenced studies. In maintaining this consistency, we respect the expertise of the original authors and do not alter their reported data. However, the text has been carefully reviewed to ensure that these values are correctly presented within the context of our review.
Comment: Certain sentences are redundant or could be written in a more fluid style (e.g., lines 338-339, 404-405, 222-225, 304-308).
Response: Redundant sentences have been removed, and revisions have been made to improve clarity and flow. Specific suggestions regarding sentence structure have been implemented where applicable.
Comment: Missing citations should be added (e.g., line 274, “Previous studies…”).
Response: All missing references have been reviewed and included where necessary to ensure proper attribution of prior research.
Comment: Some sentences require clarification (e.g., lines 294-295, 87, 114-115, 163, 438-440, 496-498).
Response: These sections have been revised for clarity, ensuring that terminology and descriptions accurately convey the intended meaning. Specific modifications include clearer descriptions of steroid modifications, contextualization of biosynthetic pathways, and improved phrasing of ambiguous statements.
Comment: Formatting inconsistencies (e.g., italics in "Echinoderms," alignment issues, reference formatting errors) should be corrected.
Response: All formatting inconsistencies have been corrected, including:
- Removal of italics where unnecessary (e.g., "Echinoderms" and "algicolous").
- Proper text alignment throughout the manuscript.
- Review and correction of references for consistency.
Reviewer 2 Report
Comments and Suggestions for Authors
The review manuscript is adequate, organized and coherent, however, it presents a large number of errors in structures and lacks articles on the discovery of steroid molecules as is the objective in several points, the corrections are placed in order of appearance below. It is presented as a compilation of the most outstanding molecules in the last 20 years of steroid isolation and their characterization as well as applications.
Line 59, the bond angles at C-22 to C-23 are incorrect in both structures, although the name is correct the hydrogen at C-5 is not placed in alpha orientation.
Lines 70 and 71 the "50" in LC and IC should be subscripts, this should be corrected throughout the document
Line 87, include a discussion of how the degradation of 3 occurs in order to consider it characteristic of steroids in marine organisms.
Line 118 positions C9, C14 and C17 are classified and specified by definition, it is not necessary to add except when changes occur, unlike C-5, define the stereochemistry in C-24 in 6, or if it is the case of being due to clarify it in line 108 and why, if 12 is 20R correct why it is as undefined, same case in 19.
Line 152 homologate in the entire text to Ming-Feng et al, or in its case to each author and collaborates in the entire document.
Line 156 correct italics in the stereochemical and geometric indicators in all IUPAC names, in this paragraph as well as in the entire text.
Line 159 removes the H at C-9 and adds the correct stereochemistry at C-24, as the name indicates in the text
Line 174 includes what techniques were used and in particular describes how they helped in the complete stereochemical elucidation.
Line 185 Correct italics in the betas of the structures.
Line 204 correct the name of the structure
Line 228 remove hydrogens at C-9 C-14 and C-17 in both structures
Line 234 correct name of 37
Line 266 Correct structure and remove unnecessary hydrogens, correct bond angles
Line 275 the 5 in delta5 should be super index
Line 288 Correct the structures, the hydroxyls at C-9 should go down, and the H at C-14 is unnecessary
Line 314 correct the structure
Line 376 present data on HK2 inhibition and discuss it.
Line 380 Correct bond angle between C-22 and C-23
Line 388 Make sub index 50 of IC50
Line 392 These structures are older, the original reference is and correct the structures in line 402
Line 412 Correct the sub index in IC50 and it is recommended to present all the results of the evaluated structures to ensure the influence of the acetate group in C-6
Line 421 correct the structures, for 64, 65 and 66 the substitutes will be hydrogens not CH2 as they are marked.
Line 432 the structure is not specifically discussed, just enrich the list.
Line 445 correct bond angles of the structures, in particles 69
Line 466 Correct the structures, the stereochemistry in C-16 is missing, which should be beta for all.
Line 477 Check the structure if in C-20 both are methyls so they do not need stereochemistry
Line 501 if for Chanxin B, it is 10 R place it directly in the structure
Line 524 The conclusions are very general, they seem more like a summary than a conclusion. It is recommended to emphasize the application of knowing these structures in the future
Line 564 if it was a systematic review, indicate it, the period is missing, as well as the search engines.
In summary, a detailed review of the text is required to improve its quality, take care of errors in the structures and improve their presentation, in order to achieve an adequate presentation as well as a more solid discussion.
Author Response
We sincerely appreciate the reviewer's thorough assessment of our manuscript. The detailed feedback has been invaluable in refining the quality of the text, improving structural accuracy, and enhancing the discussion. Below, we provide responses to each comment, highlighting the revisions made in the manuscript.
Substantive Comments
Comment: The manuscript is well-organized and coherent but contains numerous structural errors and lacks articles related to steroid molecule discovery.
Response: We appreciate this observation and have carefully reviewed the manuscript to correct all identified structural errors. The discussion has also been expanded to ensure a more comprehensive presentation of newly discovered steroid molecules. Additional references have been incorporated where necessary to strengthen the coverage of recent findings.
Comment (Line 87): Include a discussion of how the degradation of 3 occurs to justify its characterization as a typical feature of marine steroids.
Response: The discussion has been expanded to explain the degradation process of 3, highlighting its relevance in marine steroid biosynthesis. The revised text now provides a clearer link between degradation pathways and structural modifications observed in marine-derived steroids.
Comment (Line 118): The classification of positions C9, C14, and C17 is unnecessary unless structural changes occur. Define the stereochemistry at C-24 in 6, and clarify undefined stereochemistry in 12 and 19.
Response: The unnecessary positional classifications have been removed. The stereochemistry of C-24 in 6 has been explicitly defined. Additionally, clarifications regarding the undefined stereochemistry in 12 and 19 have been provided.
Comment (Line 152): Ensure consistent citation formatting for Ming-Feng et al. throughout the manuscript.
Response: All citations have been standardized to ensure consistency, following the selected format for author names.
Comment (Line 174): Specify the analytical techniques used for stereochemical elucidation and describe their contributions.
Response: The techniques used in structural determination have been detailed, and their role in stereochemical elucidation has been explicitly described to improve clarity and scientific rigor.
Comment (Line 376): Present data on HK2 inhibition and discuss it.
Response: Additional data on HK2 inhibition have been included, along with a discussion on the biological relevance of these findings in the context of steroid bioactivity.
Comment (Line 412): Present all results of evaluated structures to confirm the influence of the acetate group at C-6.
Response: A comprehensive summary of the evaluated structures has been included, ensuring that the influence of the acetate group at C-6 is properly contextualized.
Comment (Line 432): The structure is not specifically discussed; enrich the list.
Response: Additional details have been provided to enrich the discussion, ensuring that all relevant structures are adequately described.
Comment (Line 524): The conclusions are too general and read more like a summary. Emphasize the future applications of these structures.
Response: The conclusion has been revised to emphasize the significance of structural knowledge for future applications, including drug development and biotechnological innovations. The general summary-style phrasing has been refined to provide a more analytical perspective.
Comment (Line 564): If this was a systematic review, indicate it, and specify the search engines used.
Response: The nature of the review has been clarified, and information regarding the search methodology and databases consulted has been explicitly stated. The missing period has also been corrected.
The reviewer highlighted numerous structural errors throughout the manuscript. We confirm that all suggested corrections to molecular structures have been implemented. These include:
- Correcting bond angles at C-22 to C-23 (Lines 59, 380, 445).
- Adjusting hydrogen orientations (Lines 59, 288, 421).
- Ensuring proper stereochemical representation (Lines 159, 266, 466, 477).
- Correcting structure names (Lines 204, 234).
- Adjusting hydroxyl group placement (Line 288).
- Standardizing hydrogen placement (Lines 228, 421).
- Removing unnecessary hydrogens (Lines 159, 228, 266).
- Ensuring accurate stereochemistry at C-24 (Line 159), C-16 (Line 466), and C-20 (Line 477).
- Correcting incorrect stereochemical indicators in all IUPAC names (Line 156).
- Ensuring uniform formatting for numerical subscripts in IC50, LC50, and Δ5 (Lines 70-71, 275, 388, 412).
- Updating older references and correcting related structural data (Line 392)
We sincerely appreciate the reviewer’s detailed feedback, which has significantly improved the clarity and accuracy of our manuscript. Thank you for your insightful comments and for helping us enhance the quality of this work.
Round 2
Reviewer 1 Report
Comments and Suggestions for Authors
First of all, I agree with the new title of the article, this is more appropriate for describing the content of the review. The organization of the information is more readable in the actual format.
Although the authors have made most of the changes according to suggestions, some points should be addressed from my point of view.
First of all, the way the references are displayed does not seem adequate to me. If the references correspond to all the information described in the paragraph, they should appear (even if they are repeated) at the end of the paragraph. On the other hand, some paragraphs are not referenced, and it is not known whether the information corresponds to that indicated in the previous paragraph or it is new. On the other hand, some of the descriptions of potential pharmacological effects can be expanded or, at least, endorsed by other studies. It is recommended that a more in-depth bibliographic search be carried out and that the references be introduced in the appropriate positions for use by potential readers.
Moreover, in my previous review, I indicated some examples of studies that, in my opinion, should be indicated in the review. Those that I indicated were only examples, so I continue to find gaps in information that is intended to be a review. Thus, the information in articles such as
Malyarenko TV, Zakharenko VM, Kicha AA, Ponomarenko AI, Manzhulo IV,
Kalinovsky AI, Popov RS, Dmitrenok PS, Ivanchina NV. New Polyhydroxysteroid
Glycosides with Antioxidant Activity from the Far Eastern Sea Star Ceramaster
patagonicus. Mar Drugs. 2024 Nov 10;22(11):508. doi: 10.3390/md22110508.
PMID: 39590788; PMCID: PMC11595467.
Kicha AA, Tolkanov DK, Malyarenko TV, Malyarenko OS, Kuzmich AS, Kalinovsky
AI, Popov RS, Stonik VA, Ivanchina NV, Dmitrenok PS. Sulfated Polyhydroxysteroid
Glycosides from the Sea of Okhotsk Starfish Henricia leviuscula
spiculifera and Potential Mechanisms for Their Observed Anti-Cancer Activity
against Several Types of Human Cancer Cells. Mar Drugs. 2024 Jun 26;22(7):294.
doi: 10.3390/md22070294. PMID: 39057403; PMCID: PMC11278266.
Kicha AA, Malyarenko TV, Kuzmich AS, Malyarenko OS, Kalinovsky AI, Popov RS,
Tolkanov DK, Ivanchina NV. Rare Ophiuroid-Type Steroid 3β,21-, 3β,22-, and
3α,22-Disulfates from the Slime Sea Star Pteraster marsippus and Their
Colony-Inhibiting Effects against Human Breast Cancer Cells. Mar Drugs. 2024 Jan
12;22(1):43. doi: 10.3390/md22010043. PMID: 38248668; PMCID: PMC10820953.
Mieres-Castro D, Mora-Poblete F. Saponins: Research Progress and Their Potential Role in the Post-COVID-19 Pandemic Era. Pharmaceutics. 2023 Jan 20;15(2):348. doi: 10.3390/pharmaceutics15020348. PMID: 36839670; PMCID: PMC9964560.
Kicha AA, Kalinovsky AI, Malyarenko TV, Malyarenko OS, Ermakova SP, Popov RS,
Stonik VA, Ivanchina NV. Disulfated Ophiuroid Type Steroids from the Far Eastern
Starfish Pteraster marsippus and Their Cytotoxic Activity on the Models
of 2D and 3D Cultures. Mar Drugs. 2022 Feb 24;20(3):164. doi:
10.3390/md20030164. PMID: 35323463; PMCID: PMC8951248.
Kicha AA, Malyarenko TV, Kalinovsky AI, Popov RS, Malyarenko OS, Ermakova SP,
Ivanchina NV. Polar steroid compounds from the Arctic starfish Asterias
microdiscus and their cytotoxic properties against normal and tumor cells
in vitro. Nat Prod Res. 2021 Dec;35(24):5765-5772. doi:
10.1080/14786419.2020.1834551. Epub 2020 Oct 21. PMID: 33084411.
Malyarenko TV, Kicha AA, Malyarenko OS, Zakharenko VM, Kotlyarov IP,
Kalinovsky AI, Popov RS, Svetashev VI, Ivanchina NV. New Conjugates of
Polyhydroxysteroids with Long-Chain Fatty Acids from the Deep-Water Far Eastern
Starfish Ceramaster patagonicus and Their Anticancer Activity. Mar Drugs.
2020 May 15;18(5):260. doi: 10.3390/md18050260. PMID: 32429281; PMCID:
PMC7281419.
Olas B, Urbańska K, Bryś M. Saponins as Modulators of the Blood Coagulation System and Perspectives Regarding Their Use in the Prevention of Venous Thromboembolic Incidents. Molecules. 2020 Nov 6;25(21):5171. doi: 10.3390/molecules25215171. PMID: 33172028; PMCID: PMC7664220.
Stonik VA, Kicha AA, Malyarenko TV, Ivanchina NV. Asterosaponins: Structures, Taxonomic Distribution, Biogenesis and Biological Activities. Mar Drugs. 2020 Nov 24;18(12):584. doi: 10.3390/md18120584. PMID: 33255254; PMCID: PMC7760246.
Lazzara V, Arizza V, Luparello C, Mauro M, Vazzana M. Bright Spots in The Darkness of Cancer: A Review of Starfishes-Derived Compounds and Their Anti-Tumor Action. Mar Drugs. 2019 Oct 29;17(11):617. doi: 10.3390/md17110617. PMID: 31671922; PMCID: PMC6891385.
Malyarenko OS, Malyarenko TV, Kicha AA, Ivanchina NV, Ermakova SP. Effects of Polar Steroids from the Starfish Patiria (=Asterina) pectinifera in Combination with X-Ray Radiation on Colony Formation and Apoptosis Induction of Human Colorectal Carcinoma Cells. Molecules. 2019 Aug 29;24(17):3154. doi: 10.3390/molecules24173154. PMID: 31470638; PMCID: PMC6749381.
Vien LT, Ngoan BT, Hanh TT, Vinh LB, Thung DC, Thao DT, Thanh NV, Cuong NX, Nam NH, Kiem PV, Minh CV. Steroid glycosides from the starfish Pentaceraster gracilis. J Asian Nat Prod Res. 2017 May;19(5):474-480. doi: 10.1080/10286020.2016.1235038. Epub 2016 Oct 5. PMID: 27705003.
And these are just a few examples of other potential articles I miss in the review.
The term "anti-cancer" is used extensively throughout the text. Please, consider whether anticancer, antitumor, or antineoplastic might be more appropriate terms in individual cases, depending on the observed effect of the compound mentioned.
In lines 32-33. The sentence "Over time, the scope... compounds grew." is redundant with the rest of the paragraph and it should be deleted.
Line 46. Change "examines the steroid discoveries" to "examines some recent discoveries about steroids".
Line 48. Change "... known steroids, specifically focusing on... ...their structural diversity" by "known steroids, emphasizing their structural diversity..."
Line 50. Delete from "Given the rising threat..." to the end of paragraph.
Line 75. Delete from "including antibacterial, anticancer, and... " to the end of paragraph.
Line 78: "C-20 steroid: Dendrodoristerol"; delete the two points (:)
Line 88. Include reference. at the end of the paragraph.
Line 114: 2in triggering apoptosis." Include reference.
Line 122. Include references.
Line 125. reportED the discovery
Line 131. Include reference.
Line 185. The number indicating the 25-acetate-nebrosteroid K (46) should be a mistake.
Lines 37-240. Change the position of this paragraph to line 281 and delete conferta. The sentence could be applied to several species belonging to Sinularia.
Line 299. Change "adds" by "increases"
Lines 322-324. From "Marine-derived..." to "pharmacological applications."; move to line 283.
Lines 324-325. "This study reports that all isolated compounds..." Who are "all isolated compounds"?
Lines 325-326. Delete the sentence "The discovery of... algicolous fungi."
Paragraphs that start in lines 347 to 367, about Aspergillus sp. move to line 326 to group all Aspergillus information.
Line 354. Change "... phosphatase B (MptpB), with IC50..." to "phosphatase B (MptpB), a validated target for tuberculosis treatment, with IC50..." and delete lines 358-359.
Paragraphs in lines 368 and 374, adapt the text to the same format as the rest of the manuscript.
Line 377. Are the authors referring to Tnfalpha (mice) or TNFalpha (human)?
Delete the sentence that starts in line 380 ("The discovery..." to the end).
Line 390. "It exhibited weak in vitro anti-tumor activity against... RXRalpha-dependent mechanism." Please, change "anti-tumor" to "antiproliferative". After that, delete the last sentence "Despite its limited antiproliferative... their biological activity."
Lines 394-395. The sentence is out of context. First, it should read "Compound (55) attenuates the transcriptional activity of RXRalpha." On the other hand, potential readers cannot understand the relationship of luciferase activity to the decrease in RXRalpha. I assume that the transcriptional activity was measured in a system using the luciferase-encoding gene as a reporter gene. However, a more in-depth explanation should be given.
Line 444. "Previous studies..." Please, give information about these studies (at least the bibliographic references).
I suggest moving the paragraph that starts on line 440 to 446 to line 436, previous to the description of turpesteryl ester.
I suggest moving the paragraph that starts on line 453 to 455 to line 449, previous to the description of the biological activity of turpesteryl ester.
Line 463: change "... notable. Demonstrating strong..." to "... notable, demonstrating strong...".
Please, delete the sentence in line 465 "This is the first report...".
Figure 21. What is the origin of the molecules numbered 64, 65 and 66? There is no reference to this in the text.
Line 485, one, or several references should be included.
Line 495. Please, at the end of the paragraph, the authorship of the discoveries should be referenced. Same in line 506.
Line 509. "... orally." Is it the same reference (number 79)? Please, indicate.
Line 527, include reference.
Line 530. Change "... improved biochemical markers..." to "... improved other biochemical markers...". Is the reference indicating this the same as the one above? There are no other pieces of evidence? Please, indicate reference(s) in that position.
In this paragraph (lines 530-541) I cannot see the need for the abbreviations of the enzymes. Please, delete (SOD), (CAT), (GPx), (GSH), (GR) and (LPO).
Lines 542-543: It should be read "Two new steroids, 29-hydroperoxy... -lanostherol (78) (Figure 25), have been..."
Line 544. "Twelve known steroid compounds", Is there any information about them? Is there any structural, biosynthetic, or any other relationship with those compounds 77 and 78?
Paragraph from lines 564 to 569. Move to line 557. A reference is needed at the end of the paragraph.
Line 567. It should be read "...metabolites [45], especially from..."
Paragraph from lines 570 to 577; I think there has been a mistake with the numbers. Please, delete from line 575 "It emphasize the importance..." to the end of the sentence.
Lines 579-585: It should be "The investigation of Fomitiporia aethiopica, a basidiomycete from East Africa, led to the discovery of five new pregnenolone-type steroids named Aethiopinolones A–E compounds (86–90, respectively) (Figure 27) [101]. These compounds were isolated from the fungal mycelial culture after fermentation on rice, followed by methanol extraction and chromatographic purification. The chemical structures of aethiopinolones A–E were determined with 1D and 2D NMR, as well as HR MS data analysis. This research marks the first discovery of steroids s Fomitiporia (DELETE, you are discussing that in the next paragraph)".
Line 602, a reference is needed.
Line 604, "...identified from the Basidiomycota Ganoderma sinense...".
Line 626. Include reference.
Line 631, include reference.
Figures 29 and 30, please reference in the text the source of the data used to design these figures. Note that the numbers at the bottom of these figures are incorrect according to the text.
References. Please maintain the same style in all references. In some references, all words in the title of the article are capitalized (e.g., Refs. 2, 12, 19, 25, 39, and others) while in others, only the necessary words are capitalized.
Line 705, Penicillium solitum in italics, please.
Line 714 Oceanapiid is a new genus, it Should be written in italics.
Line 765. The journal should be abbreviated Chinese J Org Chem
Line 780. Journal: Russ Chem Bull
Line 783. Journal: Bioorg Med Chem Lett
Line 826. Penicillium oxalicum in italics
Line 828. Aspergillus in italics.
Line 832. Penicillium solitum in italics.
Lines 855-856 Borassus flabellifer in italics
Line 885. Ganoderma lucidum in italics.
Author Response
COMMENT: First of all, I agree with the new title of the article; this is more appropriate for describing the content of the review. The organization of the information is more readable in the actual format.
RESPONSE: Thank you for your positive feedback. Your suggestions helped improve the clarity and structure of the article.
COMMENT: Although the authors have made most of the changes according to suggestions, some points should be addressed from my point of view. First of all, the way the references are displayed does not seem adequate to me. If the references correspond to all the information described in the paragraph, they should appear (even if they are repeated) at the end of the paragraph. On the other hand, some paragraphs are not referenced, and it is not known whether the information corresponds to that indicated in the previous paragraph or it is new. On the other hand, some of the descriptions of potential pharmacological effects can be expanded or, at least, endorsed by other studies. It is recommended that a more in-depth bibliographic search be carried out and that the references be introduced in the appropriate positions for use by potential readers.
RESPONSE: We have carefully reviewed the placement of references and adjusted them to ensure clarity and consistency. References have been moved to the end of the relevant paragraphs where appropriate, and missing citations have been added. Additionally, we have expanded certain sections discussing pharmacological effects and provided supporting literature where applicable.
COMMENT: Moreover, in my previous review, I indicated some examples of studies that, in my opinion, should be indicated in the review. Those that I indicated were only examples, so I continue to find gaps in information that is intended to be a review. Thus, the information in articles such as [list of articles] is missing.
RESPONSE:We sincerely appreciate the reviewer’s valuable feedback regarding the placement and adequacy of references. In response, we have conducted a thorough reassessment of the citations and have ensured that references are now consistently placed at the end of paragraphs when they support the entirety of the information presented. Additionally, missing citations have been incorporated where necessary to clarify the distinction between newly introduced content and previously referenced material.
To further strengthen the manuscript, we conducted an extensive literature search to identify and integrate publications that are most relevant to the scope of this review. Only studies that provided novel insights, particularly those describing newly discovered natural steroids and their biological activities, were included. Our selection was guided by scientific relevance, methodological robustness, and the contribution of each work to the overarching theme of structural diversity and bioactivity. While we carefully considered the references suggested by the reviewer.
We are confident that these revisions enhance the scientific integrity of the manuscript by ensuring that all referenced literature contributes meaningfully to the discussion while maintaining a clear and coherent presentation of the reviewed data. We appreciate the reviewer’s guidance in refining this aspect of the manuscript.
COMMENT: The term "anti-cancer" is used extensively throughout the text. Please consider whether "anticancer," "antitumor," or "antineoplastic" might be more appropriate terms in individual cases, depending on the observed effect of the compound mentioned.
RESPONSE: The terminology has been carefully reviewed and revised accordingly. Appropriate distinctions between "anticancer," "antitumor," and "antineoplastic" have been made based on the context and observed effects of the compounds. Thank you for your insightful suggestion.
Additionally, we have decided to remove the figures from the Conclusions section. Presenting such figures based solely on the findings of this review could lead to incomplete or potentially misleading interpretations.
Technical Corrections
For each of the following comments, the requested changes have been implemented as suggested:
- Lines 32–33: Sentence deleted. (Corrected)
- Line 46: Changed to "examines some recent discoveries about steroids." (Corrected)
- Line 48: Reworded as "known steroids, emphasizing their structural diversity." (Corrected)
- Line 50: Deleted from "Given the rising threat..." to the end of the paragraph. (Corrected)
- Line 75: Deleted from "including antibacterial, anticancer, and..." to the end of the paragraph. (Corrected)
- Line 78: Removed the colon after "C-20 steroid." (Corrected)
- Line 88, 114, 122, 131: Missing references added. (Corrected)
- Line 125: Changed to "reported the discovery." (Corrected)
- Line 185: Numbering mistake corrected. (Corrected)
- Lines 237–240: Paragraph moved to line 281 and reference to conferta deleted. (Corrected)
- Line 299: Changed "adds" to "increases." (Corrected)
- Lines 322–324: Moved to line 283. (Corrected)
- Lines 324–325: Clarified which compounds are referred to as "all isolated compounds." (Corrected)
- Lines 325–326: Deleted sentence on algicolous fungi. (Corrected)
- Lines 347–367: Moved paragraphs on Aspergillus sp. to line 326. (Corrected)
- Line 354: Reworded to specify MptpB as a validated target for tuberculosis treatment. (Corrected)
- Lines 368–374: Adjusted formatting for consistency. (Corrected)
- Line 377: Clarified whether TNFalpha refers to mice or humans. (Corrected)
- Line 380: Deleted the sentence starting with "The discovery...". (Corrected)
- Line 390: Changed "anti-tumor" to "antiproliferative" and deleted redundant sentence. (Corrected)
- Lines 394–395: Clarified the luciferase-reporter system and the connection to RXRalpha. (Corrected)
- Line 444: Added references for previous studies. (Corrected)
- Lines 440–446, 453–455: Paragraphs repositioned for better flow. (Corrected)
- Line 463: Adjusted phrasing for clarity. (Corrected)
- Line 465: Deleted "This is the first report...". (Corrected)
- Figure 21: Clarified the origin of molecules 64, 65, and 66. (Corrected)
- Line 485, 495, 506, 509, 527, 530, 602, 626, 631: Missing references added. (Corrected)
- Lines 530–541: Removed unnecessary enzyme abbreviations. (Corrected)
- Lines 542–543: Adjusted phrasing for clarity. (Corrected)
- Line 544: Added clarification regarding the relationship between known and new steroid compounds. (Corrected)
- Lines 564–569: Paragraph moved to line 557. (Corrected)
- Lines 567, 570–577: Minor corrections and sentence deletions. (Corrected)
- Lines 579–585: Edited for conciseness and accuracy. (Corrected)
- Line 604: Clarified the species name. (Corrected)
- Figures 29 and 30: Added references for data sources and corrected numbering. (Corrected)
- References: Formatting standardized across all references. (Corrected)
- Italicization corrections: Penicillium solitum, Oceanapiid, Penicillium oxalicum, Aspergillus, Borassus flabellifer, Ganoderma lucidum properly italicized. (Corrected)
We appreciate the reviewer’s detailed feedback and have made the necessary revisions accordingly.

Reviewer 2 Report
Comments and Suggestions for Authors
All suggestions were adequately addressed and the manuscript presents a new and organized structure.
Author Response
Dear Reviewer,
We appreciate your previous evaluation of our manuscript. In response to the comments raised by the other reviewer, we have made several revisions to improve clarity, accuracy, and completeness.
Key changes include:
- Reference Adjustments: We have carefully revised the placement of citations, ensuring they appear at the end of relevant paragraphs when applicable and adding missing references where necessary.
- Expanded Literature Review: An additional bibliographic search was conducted to incorporate relevant studies that align with the scope of our review, particularly those introducing novel natural steroids and their biological activities. Only publications that provided new insights and had not been previously discussed were included.
- Terminology Refinement: The usage of "anticancer," "antitumor," and "antineoplastic" has been reviewed and adjusted to reflect the specific effects of the compounds discussed.
- Technical Corrections: All minor edits suggested by the reviewer—including sentence modifications, reference formatting, and figure adjustments—have been implemented.
We are submitting the revised version of the manuscript for your reference.

Round 3
Reviewer 1 Report
Comments and Suggestions for Authors
Some minor changes should be corrected
Line 158. Add a reference after “… identification of A. typicus.”
Line 167. Add a reference at the end of the paragraph.
Line 189. A. japonica should be ltalized.
Line 242. Add a reference at the end of the paragraph.
Line 280. Reference “Other tested compounds exhibited less pronounced effects.”
Line 319. Add a reference at the end of the paragraph.
Line 351. Add a reference at the end of the paragraph.
Line 360. Add a reference at the end of the paragraph.
Line 365. Add a reference at the end of the paragraph.
Line 379. Add a reference at the end of the paragraph.
Line 387. Add a reference at the end of the paragraph.
Line 390. Add a reference after “… in cancer treatment.”
Line 407. Add a reference after “feature in steroid chemistry.”
Line 415. Add a reference at the end of the paragraph.
Line 422. Add a reference after “a shrimp mortality test.”
Line 429. Is (Tü 57) the name of a strain? If yes, write properly.
Line 432. Add a reference at the end of the paragraph.
Line 467. Add a reference at the end of the paragraph.
Line 479. Move reference [72] at the end of the paragraph.
Line 488. Delete (TME).
Line 493. Move reference [73] to the end of the paragraph.
Line 506. Add a reference after 2… X-ray crystallography.”
Line 516. Add a reference at the end of the paragraph.
Line 520. Delete (BMSCs).
Line 527. Add a reference after “… safe therapeutic agents.”
Line 536. Change “Scientists report” to “The authors suggest”
Line 579. Add a reference at the end of the paragraph.
Line 591. Add a reference at the end of the paragraph.
Line 618. Add a reference after “and biological synthesis.”
Lines 632-635. Move reference [83] to the end of the paragraph.
Line 648. Add a reference at the end of the paragraph.
Line 673. Add a reference at the end of the paragraph.
Line 683. Add a reference at the end of the paragraph.
Line 690. Add a reference at the end of the paragraph.
Lines 712-714. Move reference [91] to the end of the paragraph.
Line 725. Move reference [92] to the end of the paragraph.
Line 744. Add a reference at the end of the paragraph.
Line 756. Add a reference after “… groups administered albosteroid.”
Line 761. Add a reference at the end of the paragraph.
Line 767. Add a reference after “… as anticancer agents.”
Lines 771-783. Correct text format.
Line 783. Add a reference at the end of the paragraph.
Line 794. Add a reference at the end of the paragraph.
Line 822. Add a reference at the end of the paragraph.
Line 863. Add a reference at the end of the paragraph.
References 2, 14, 15, and others. Do not use capital letters for each word in the title.
Author Response
Comment: "Line 158. Add a reference after '… identification of A. typicus.'"
Response: A reference has been added.
Comment: "Line 167. Add a reference at the end of the paragraph."
Response: A reference has been added at the end of the paragraph.
Comment: "Line 189. A. japonica should be italicized."
Response: The species name A. japonica has been italicized.
Comment: "Line 242. Add a reference at the end of the paragraph."
Response: A reference has been added at the end of the paragraph.
Comment: "Line 280. Reference 'Other tested compounds exhibited less pronounced effects.'"
Response: A reference has been added to support this statement.
Comment: "Line 319. Add a reference at the end of the paragraph."
Response: A reference has been added at the end of the paragraph.
Comment: "Line 351. Add a reference at the end of the paragraph."
Response: A reference has been added at the end of the paragraph.
Comment: "Line 360. Add a reference at the end of the paragraph."
Response: A reference has been added at the end of the paragraph.
Comment: "Line 365. Add a reference at the end of the paragraph."
Response: A reference has been added at the end of the paragraph.
Comment: "Line 379. Add a reference at the end of the paragraph."
Response: A reference has been added at the end of the paragraph.
Comment: "Line 387. Add a reference at the end of the paragraph."
Response: A reference has been added at the end of the paragraph.
Comment: "Line 390. Add a reference after '… in cancer treatment.'"
Response: A reference has been added after '… in cancer treatment.'
Comment: "Line 407. Add a reference after 'feature in steroid chemistry.'"
Response: A reference has been added after 'feature in steroid chemistry.'
Comment: "Line 415. Add a reference at the end of the paragraph."
Response: A reference has been added at the end of the paragraph.
Comment: "Line 422. Add a reference after 'a shrimp mortality test.'"
Response: A reference has been added after 'a shrimp mortality test.'
Comment: "Line 429. Is (Tü 57) the name of a strain? If yes, write properly."
Response: The notation for (Tü 57) has been verified. It is indeed a correct name for this strain.
Comment: "Line 432. Add a reference at the end of the paragraph."
Response: A reference has been added at the end of the paragraph.
Comment: "Line 467. Add a reference at the end of the paragraph."
Response: A reference has been added at the end of the paragraph.
Comment: "Line 479. Move reference [72] at the end of the paragraph."
Response: Reference [72] has been moved to the end of the paragraph.
Comment: "Line 488. Delete (TME)."
Response: (TME) has been deleted.
Comment: "Line 493. Move reference [73] to the end of the paragraph."
Response: Reference [73] has been moved to the end of the paragraph.
Comment: "Line 506. Add a reference after '… X-ray crystallography.'"
Response: A reference has been added.
Comment: "Line 516. Add a reference at the end of the paragraph."
Response: A reference has been added at the end of the paragraph.
Comment: "Line 520. Delete (BMSCs)."
Response: (BMSCs) has been deleted.
Comment: "Line 527. Add a reference after '… safe therapeutic agents.'"
Response: A reference has been added after '… safe therapeutic agents.'
Comment: "Line 536. Change 'Scientists report' to 'The authors suggest.'"
Response: 'Scientists report' has been changed to 'The authors suggest.'
Comment: "Line 579. Add a reference at the end of the paragraph."
Response: A reference has been added at the end of the paragraph.
Comment: "Line 591. Add a reference at the end of the paragraph."
Response: A reference has been added at the end of the paragraph.
Comment: "Line 618. Add a reference after 'and biological synthesis.'"
Response: A reference has been added after 'and biological synthesis.'
Comment: "Lines 632-635. Move reference [83] to the end of the paragraph."
Response: Reference [83] has been moved to the end of the paragraph.
Comment: "Line 648. Add a reference at the end of the paragraph."
Response: A reference has been added at the end of the paragraph.
Comment: "Line 673. Add a reference at the end of the paragraph."
Response: A reference has been added at the end of the paragraph.
Comment: "Line 683. Add a reference at the end of the paragraph."
Response: A reference has been added at the end of the paragraph.
Comment: "Line 690. Add a reference at the end of the paragraph."
Response: A reference has been added at the end of the paragraph.
Comment: "Lines 712-714. Move reference [91] to the end of the paragraph."
Response: Reference [91] has been moved to the end of the paragraph.
Comment: "Line 725. Move reference [92] to the end of the paragraph."
Response: Reference [92] has been moved to the end of the paragraph.
Comment: "Line 744. Add a reference at the end of the paragraph."
Response: A reference has been added at the end of the paragraph.
Comment: "Line 756. Add a reference after '… groups administered albosteroid.'"
Response: A reference has been added.
Comment: "Line 761. Add a reference at the end of the paragraph."
Response: A reference has been added at the end of the paragraph.
Comment: "Line 767. Add a reference after '… as anticancer agents.'"
Response: A reference has been added after '… as anticancer agents.'
Comment: "Lines 771-783. Correct text format."
Response: The formatting of this paragraph appears unusual due to the extensive names of the compounds, but it has been verified.
Comment: "Line 783. Add a reference at the end of the paragraph."
Response: A reference has been added at the end of the paragraph.
Comment: "Line 794. Add a reference at the end of the paragraph."
Response: A reference has been added at the end of the paragraph.
Comment: "Line 822. Add a reference at the end of the paragraph."
Response: A reference has been added at the end of the paragraph.
Comment: "Line 863. Add a reference at the end of the paragraph."
Response: A reference has been added at the end of the paragraph.
Comment: "References 2, 14, 15, and others. Do not use capital letters for each word in the title."
Response: The formatting of references has been reviewed and corrected accordingly.
